# Blockade of IGF2R improves muscle regeneration and ameliorates Duchenne muscular dystrophy

Pamela Bella[1], Andrea Farini[1], Stefania Banfi[2], Daniele Parolini[3], Noemi Tonna[4], Mirella Meregalli[1], Marzia Belicchi[1], Silvia Erratico[5], Pasqualina D'Ursi[6], Fabio Bianco[4], Mariella Legato[1], Chiara Ruocco[7], Clementina Sitzia[8], Simone Sangiorgi[9], Chiara Villa[1], Giuseppe D'Antona[10], Luciano Milanesi[6], Enzo Nisoli[7], PierLuigi Mauri[6] & Yvan Torrente[1,*] (iD)

## Abstract

Duchenne muscular dystrophy (DMD) is a debilitating fatal X-linked muscle disorder. Recent findings indicate that IGFs play a central role in skeletal muscle regeneration and development. Among IGFs, insulinlike growth factor 2 (IGF2) is a key regulator of cell growth, survival, migration and differentiation. The type 2 IGF receptor (IGF2R) modulates circulating and tissue levels of IGF2 by targeting it to lysosomes for degradation. We found that IGF2R and the store-operated $Ca^{2+}$ channel CD20 share a common hydrophobic binding motif that stabilizes their association. Silencing CD20 decreased myoblast differentiation, whereas blockade of IGF2R increased proliferation and differentiation in myoblasts via the calmodulin/calcineurin/NFAT pathway. Remarkably, anti-IGF2R induced CD20 phosphorylation, leading to the activation of sarcoplasmic/endoplasmic reticulum $Ca^{2+}$-ATPase (SERCA) and removal of intracellular $Ca^{2+}$. Interestingly, we found that IGF2R expression was increased in dystrophic skeletal muscle of human DMD patients and *mdx* mice. Blockade of IGF2R by neutralizing antibodies stimulated muscle regeneration, induced force recovery and normalized capillary architecture in dystrophic *mdx* mice representing an encouraging starting point for the development of new biological therapies for DMD.

**Keywords** DMD; IGF2; IGF2R; muscle regeneration; muscular dystrophy
**Subject Categories** Musculoskeletal System; Pharmacology & Drug Discovery

## Introduction

Duchenne muscular dystrophy (DMD) is a devastating X-linked disease characterized by progressive muscle weakness and caused by a lack of dystrophin protein in the sarcolemma of muscle fibres (Emery, 2002). Impaired muscle regeneration with exhaustion of the satellite cell pool is a major hallmark of DMD. Members of the insulin-like growth factor (IGF) family are secreted during muscle repair and promote muscle regeneration and hypertrophy. Among the IGFs, IGF1 signalling has been extensively characterized for its capacity to promote the proliferation and differentiation of satellite cells, regulate muscle hypertrophy and ameliorate the features of muscular dystrophy (Florini *et al*, 1996; Barton *et al*, 2002; Zanou & Gailly, 2013). Nevertheless, little is known about the role of IGF2 in skeletal muscle development and regeneration *in vivo*. *In vitro* studies have shown that the IGF2 protein plays a role in a later step of myoblast differentiation (Florini *et al*, 1991; Wilson *et al*, 2003; Ge *et al*, 2011). Interestingly, it was previously shown that there is a link between the *Myod* and *Igf2* genes in myoblast cell culture (Montarras *et al*, 2005). Further studies suggested that IGF2, by binding to the IGF1 receptor, activates the Akt pathway and downstream targets of *Myod*, although the exact mechanisms underlying these processes have not been identified (Wilson & Rotwein, 2006, 2007). IGF2 signalling is regulated by IGF-binding proteins, which sequester circulating IGF2; the IGF2 receptor (IGF2R), which reduces IGF2 bioactivity (Brown *et al*, 2009); and the insulin receptor and IGF1 receptor, both of which can be activated by IGF2 (Livingstone, 2013). The extracytoplasmic region of IGF2R has three binding sites: one for IGF2 in domain 11 and two for Man-6-P in domains 3, 5 and 9 (Dahms *et al*, 1993; Reddy *et al*, 2004; Williams

1   Stem Cell Laboratory, Department of Pathophysiology and Transplantation, Unit of Neurology, Fondazione IRCCS Ca' Granda Ospedale Maggiore Policlinico, Centro Dino Ferrari, Universitá degli Studi di Milano, Milan, Italy
2   Hematology Department Fondazione IRCCS, Department of Oncology and Hemato-oncology, Istituto Nazionale dei Tumori, Universitá degli Studi di Milano, Milan, Italy
3   Thermo Fisher Scientific, Life Technologies Italia, Monza, Italy
4   Neuro-Zone s.r.l., Open Zone, Milano, Italy
5   Novystem Spa, Milan, Italy
6   Institute of Technologies in Biomedicine, National Research Council (ITB-CNR), Milan, Italy
7   Department of Medical Biotechnology and Translational Medicine, Center for Study and Research on Obesity, Milan University, Milan, Italy
8   UOC SMEL-1, Scuola di Specializzazione di Patologia Clinica e Biochimica Clinica, Università degli Studi di Milano, Milan, Italy
9   Neurosurgery Unit, Department of Surgery, ASST Lariana-S. Anna Hospital, Como, Italy
10  Department of Public Health, Experimental and Forensic Medicine, Pavia University, Pavia, Italy
   *Corresponding author. Tel: +39 0255 033874; E-mail: yvan.torrente@unimi.it

et al, 2007). Binding between IGF2 and IGF2R induces the lysosomal degradation of IGF2. Hence, IGF2R serves to clear IGF2 from the circulation or degrade excess circulating IGF2 (Fargeas et al, 2003; Spicer & Aad, 2007). Soluble IGF2R can affect the size of some organs exclusively by reducing the biological activity of IGF2 (Zaina & Squire, 1998).

Here, we report that IGF2R and the store-operated $Ca^{2+}$ channel CD20 share a common hydrophobic binding motif that stabilizes their association. Intracellular $Ca^{2+}$ regulation is compromised in dystrophic muscle fibres. Mechanisms that affect the influx of $Ca^{2+}$ into dystrophic muscle fibres include membrane tears (Turner et al, 1988; Straub et al, 1997; Blake et al, 2002), stretch-activated channels (Gervasio et al, 2008), $Ca^{2+}$ leak channels (Fong et al, 1990) and leaky $Ca^{2+}$ release channels (Bellinger et al, 2009); it has also been speculated that the function of SERCA, the main protein responsible for $Ca^{2+}$ reuptake into the sarcoplasmic reticulum (SR), is compromised in mdx mice (Tutdibi et al, 1999; Nicolas-Metral et al, 2001).

We found that IGF2R targeting regulated the phosphorylation of CD20 and thereby induced SERCA activation in myoblasts. Interestingly, a delay in muscle differentiation was observed in CD20-silenced (shCD20) myoblasts, whereas the expression levels of early and late differentiation markers were increased after blockade of IGF2R. These features were accompanied by the activation of the calmodulin/calcineurin/NFAT pathway, suggesting that an IGF post-translational modulatory mechanism regulates muscle differentiation. Notably, IGF2R expression was increased in the dystrophic muscle tissues of mdx mice, while the phosphorylation of IGF2R was significantly decreased. Because IGF2R and CD20 interactions could affect dystrophic muscle tissues, we hypothesized that IGF clearance was faster and its bioavailability lower in dystrophic muscles than in normal muscles and that these changes were accompanied by perturbation of $Ca^{2+}$ reuptake into the SR. Remarkably, in mdx mice, blockade of IGF2R increased muscle regeneration and significantly recovered muscle force via SERCA activation and $Ca^{2+}$ reuptake. The IGF2 pathway affects vascular architecture, and the vessel structures of dystrophic skeletal muscles were clearly disorganized in mdx mice; hence, we examined the effect of anti-IGF2R antibodies on blood vessels in the skeletal muscles of mdx mice and found that muscle capillaries were linearized and exhibited normal architecture and maturation. Overall, these data demonstrated that a biological therapy targeting IGF2R leads to improvement of muscle regeneration and suppression of the pathological cascade associated with muscle dystrophic events.

# Results

## CD20 phosphorylation is affected by IGF-driven pathway

Given the finding that CD20 acts as a mediator/modulator of store-operated calcium entry (SOCE) in skeletal myoblasts (Parolini et al, 2012), we were prompted to evaluate the functional impact of CD20 in C2C12 myoblast differentiation and analyse its possible interactions with the IGF pathway. IGF1 and IGF1R expression were not increased in 10 nM IGF1-treated C2C12 myoblasts (Fig 1A), but IGF1 treatment did induce a significant increase in transcription-dependent IGF2 production (Fig 1A). Immunofluorescence staining

revealed that treating C2C12 cells with 10 nM IGF1 increased IGF2 production (Fig 1B). Moreover, over-expressing CD20 in C2C12 cells also resulted in a transcriptional increase in IGF2 expression (Fig 1A). In contrast, IGF2 expression was not detected in untreated C2C12 cells (Fig 1A and B) or CD20 shRNA-transfected C2C12 cells (Fig 1A). To assess the effect of stable CD20 inhibition, C2C12 cells were exposed to lentiviral particles designed to deliver constructs encoding shRNAs that targeted the CD20 mRNA. Knockdown efficiency was assayed after infected cells were selected by WB analysis, which revealed that CD20 protein expression was 60% silenced (Fig EV1A). To assess whether CD20 phosphorylation is affected by an IGF1-driven pathway, C2C12 myoblasts were exposed to 1 nM or 10 nM IGF1 for 2 h or overnight. The impact on CD20 phosphorylation was then detected by specific phosphor-Ser and phosphor-Thr antibodies. After 2 h of exposure to 10 nM IGF1, the level of phosphorylation of CD20 was significantly increased at both Ser and Thr residues (Fig 1C). CD20 serine phosphorylation was also increased after overnight exposure to 1 nM and 10 nM IGF1, although to a lesser extent (Fig 1C). Blockade of IGF2R induced a significant higher level of CD20 phosphorylation than was observed in IGF1-, shCD20- and anti-Flag-treated myoblasts (Fig 1D). Interestingly, CD20 serine phosphorylation was significantly increased in myoblasts co-stimulated with either 10 or 100 nM of IGF2 (Fig 1E). Together, these findings indicate that the activation of CD20-related signalling can be induced by IGFs in skeletal myoblasts. Moreover, IGF2R expression and phosphorylation were reduced in shCD20- and anti-IGF2R-treated myoblasts (Fig 1F). Treatment with anti-IGF2R increased IGF2R-Gαi2 interactions and regulated IGF1R phosphorylation, suggesting a shift in IGF1-IGF1R interactions (Fig 1G and H).

## IGF2R binding to CD20 is relevant for myogenic differentiation

We next sought to verify whether there are interactions between IGF2R and CD20. Automated protein docking of the transmembrane model of IGF2R with the crystal structure of CD20 predicts the binding pocket and identifies the residue critical for the binding. Structure of IGF2R with IGF2 compared to docking poses of epitope 3 shows that CD20 binds IGF2R domain 11 in the IGF2 binding site. This binding site region consists of a hydrophobic pocket centred on the CD loop, surrounded by polar and charged residues in the AB, EF and HI loops that complement surface charge on IGF2. The helical region (residues 178–184) of CD20 mediates the binding to receptor as well as IGF2 (Fig 2A). Indeed, Tyr18 of CD20 acts as an anchor making contact with the hydrophobic cluster Tyr1542, Phe 1567 and Leu 1629 of IGF2R domain 11 (Fig 2B), while for IGF2 the Phe19 of IGF2 is involved in the hydrophobic interaction (Fig 2C). Site-directed mutagenesis and structural studies have been shown that Phe 19 is important for IGF2R binding (Chuprin et al, 2015).

CD20 (Swiss-Prot code: P11836) is an integral membrane protein with intracellular N- and C-termini, four transmembrane spans (TM1–4) and an extracellular domain located between TM3 and TM4. Its only extracellular portions are two short loops located at positions 72–80 and 142–182. Mutagenesis and epitope mapping studies identified two CD20 epitopes in these extracellular domains. Epitope 2 spans residues 146–160, while epitope 3 spans residues 168–175. The X-ray structures of the Fab fragment in the complex with epitope 3 mimicked the large extracellular loop of CD20. The

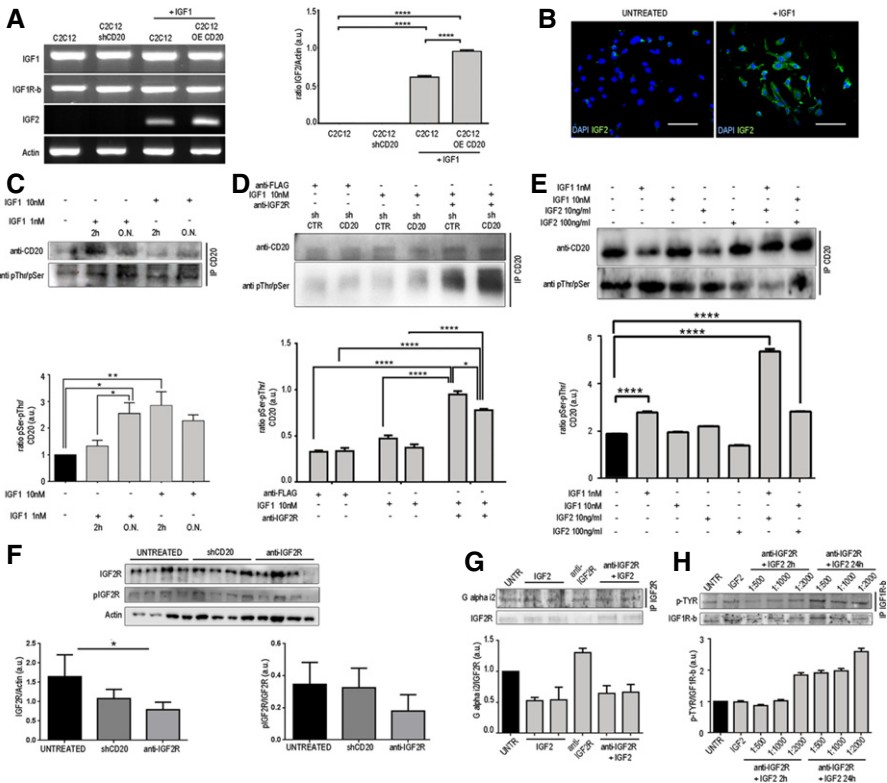

**Figure 1. IGF2R blockade results in CD20 phosphorylation.**

A    RT–PCR expression and quantification of IGF1, IGF1Rβ and IGF2 levels in untreated C2C12 and shCD20 C2C12 myoblasts and 10 nM IGF1-treated C2C12 and over-expressing CD20 C2C12 myoblasts. Each experiment was replicated independently four times. Two-way ANOVA. ****$P < 0.0001$. All values are expressed as the mean ± SEM.

B    Immunofluorescence for IGF2 (in green) in untreated and 10 nM IGF1-treated C2C12 myoblasts. Scale bars = 75 μm.

C–E   Representative CD20 immunoprecipitation using anti-pSer + pThr in (C) C2C12 cells treated with IGF1 for 2 h or overnight (ON), (D) sh-empty (shCTR)- and shCD20-treated C2C12 cells treated with anti-Flag and 10 nM anti-IGF2R, (E) C2C12 cells treated with IGF1 and IGF2, as indicated. Densitometric analysis of data is expressed as the ratio of CD20/vinculin or pSer + pThr/CD20 and is shown normalized to vinculin in arbitrary units in the lower panels. Two-way ANOVA. *$P < 0.05$; **$P < 0.01$; ****$P < 0.0001$. Each experiment was performed in triplicate wells. All values are expressed as the mean ± SEM.

F    Representative WB of IGF2R and phosphorylated IGF2R (pIGF2R) in untreated, shCD20-treated and anti-IGF2R-treated C2C12 cells. Densitometric analysis of data is expressed as the ratio of IGF2R/actin or pIGF2R/IGF2R in arbitrary units in the lower panels. One-way ANOVA. *$P < 0.05$. Each experiment was performed in triplicate wells. All values are expressed as the mean ± SEM.

G    IGF2R immunoprecipitation products were immunoblotted for Gαi2 and WB expression of IGF2R in untreated, IGF2-treated, anti-IGF2R-treated and IGF2 + anti-IGF2R-treated C2C12 cells. Densitometric analysis of data is expressed as the ratio of Gαi2/IGF2R in arbitrary units in the lower panel. Each experiment was performed in triplicate wells. All values are expressed as the mean ± SEM.

H    Representative WB of IGF1Rβ and IGF1Rβ immunoprecipitation products immunoblotted for pTyr in untreated, IGF2-treated and IGF2 + anti-IGF2R-treated (1:500, 1:1,000 and 1:2,000 dilutions of anti-IGF2R) C2C12 cells (cells were treated for 2 and 24 h). Densitometric analysis of data is expressed as the ratio of pTYR/IGF1Rβ in arbitrary units in the lower panel. Each experiment was performed in triplicate wells. All values are expressed as the mean ± SEM.

Source data are available online for this figure.

complex structure of human IGF2R domains 11–13, which bind to IGF2 (PDB code 2v5p), is also available. Using a protein–protein docking procedure, we evaluated the IGF2R-CD20 epitope 3 interaction using the ClusPro server. Four initial CD20 epitope 3 conformations (obtained with the PDB codes 2osl, 3bky and 3pp4) and one IGF2R conformation were used as the starting structures for the docking simulations. Computational steps were performed as follows: (i) rigid-body docking by sampling billions of conformations, (ii) RMSD-based clustering of the 1,000 structures with the lowest energy to find highly populated clusters that will represent the most likely models of the complex and (iii) refinement of the selected structures using energy minimization. For each simulation, ClusPro returned ten clusters with low energy, and a representative

pose was chosen from the most populated cluster with the lowest energy on visual inspection (Fig 2C). In co-immunoprecipitation experiments, the IGF2R protein formed a stable complex with CD20 (Fig 2E), and shCD20 abolished or decreased the ability of IGF2R to interact with CD20 (Fig 2E). These findings suggest that IGF2R and CD20 share a common hydrophobic binding motif that stabilizes their association. Moreover, over-expression of CD20 increased the expression of IGF2R in C2C12 myoblasts (Fig 2D). These results verify that functional interactions occur between CD20 and IGF2R during myoblast differentiation. The expression of early and late differentiation myogenic markers demonstrated that myoblasts started to express myogenin after 2 days in differentiation medium (DM) and that its expression peaked after 4 days before decreasing after

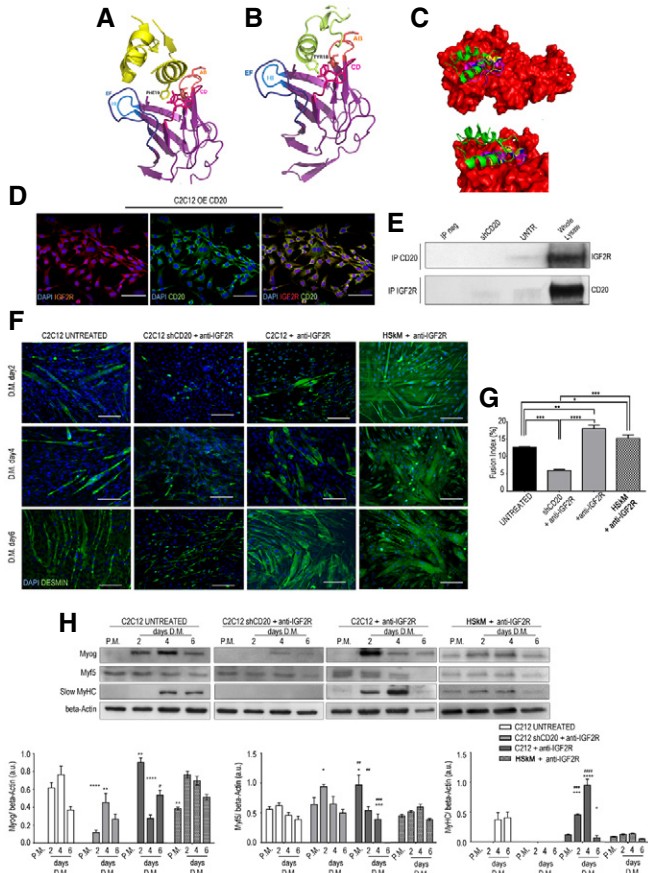

**Figure 2. IGF2R binding to CD20 is relevant for myogenic differentiation.**

A–C  Bioinformatic prediction of IGF2R and CD20 cross-reactivity. Cartoon representation of the interaction of IGF2 (yellow) and IGF2R domain 11 complex from X-ray structure (PDB code 2v5p). IGF2R domain 11 AB, CD, EF and HI loops and residues (shown in sticks format) involved in the hydrophobic interactions are shown (A). Cartoon representation of the IGF2R domain 11–CD20 (lime) complex obtained from docking simulations and residues involved in hydrophobic interactions is shown in stick format (B). The structure of IGF2R (in red) and IGF2 (green) is shown compared to docking poses of epitope 3. These data show that the helical region (residues 178–184) of the epitope mediates binding to the receptor as well as IGF2. The C-terminal helix of epitope 3 partially overlaps the first α-helix of IGF2. ClusPro-dock IGF2R-CD20 binding epitope poses corresponding to PDB codes 2v5p-2oslP, 2v5p-2oslQ and 2v5p-3pp4P are coloured in yellow, blue and cyan, respectively (C).

D  Over-expressing CD20 in C2C12 myoblasts that co-expressed CD20 (in green) and IGF2R (in red). DAPI-labelled nuclei are shown in blue. Scale bars = 75 μm.

E  Representative CD20 and IGF2R immunoprecipitation products immunoblotted for IGF2R and CD20, respectively, in untreated and shCD20-treated C2C12 cell membranes and whole lysates of proteins. The immunoprecipitation output is shown as IP neg.

F  Myotube immunofluorescence of cells in proliferation medium (P.M.) and after 2, 4 and 6 days of myogenic differentiation in serum-free medium. Control (untreated), shCD20-treated C2C12 cells, C2C12 and HSkM myoblasts pre-treated with anti-IGF2R for 24 h were stained. Desmin-positive myotubes are shown in green. Scale bars = 75 μm.

G  Fusion index quantification after 6 days of differentiation. One-way ANOVA. *$P < 0.05$; **$P < 0.01$; ***$P < 0.001$; ****$P < 0.0001$. Each experiment was performed in triplicate wells. All values are expressed as mean ± SEM.

H  Representative WB of anti-myogenin, anti-Myf5, anti-MyHC and anti-β-actin in total protein lysates obtained from untreated, shCD20-treated C2C12 cells, and C2C12 and HSkM myoblasts pre-treated with anti-IGF2R for 24 h; cells were collected under P.M. and after 2, 4 and 6 days of myogenic differentiation. Densitometric analysis of WB data expressed as the ratio of the indicated antibody/β-actin in arbitrary units. Two-way ANOVA test. *$P < 0.05$; **$P < 0.01$; ***$P < 0.001$; ****$P < 0.0001$ in comparison with the results obtained in untreated cells at the corresponding time point. ##$P < 0.01$; ###$P < 0.001$; ####$P < 0.0001$ in comparison with the results obtained in shCD20-treated C2C12 cells at the corresponding time point. Each experiment was performed in triplicate wells. All values are expressed as the mean ± SEM.

Source data are available online for this figure.

6 days of differentiation (Fig 2F–H). Accordingly, the expression of MyHC was first observed on day 4 and persisted at day 6. Myoblasts exposed to blockade of IGF2R showed a significant increase in the expression of early and late differentiation markers. Myogenin expression was significantly higher at day 2 in treated than in untreated cells, as was MyHC expression, which was detectable at the same time points, peaked at 4 days and then decreased at day 6 (Fig 2G and H). Surprisingly, treatment with shCD20 resulted in a consistent delay in myoblast differentiation after IFG2R blockade: myogenin expression was undetectable after 2 days of differentiation and was only slightly detectable on day 4, whereas MyHC was not expressed at any of the evaluated time points (Fig 2G and H). Fusion index analysis confirmed the loss of differentiation potential in shCD20-treated C2C12 myoblasts exposed to IGF2R blockade (Fig 2G). Moreover, IGF2R blockade-induced C2C12 myotubes were found to be longer and to have a low number of nuclei per fibre compared to untreated C2C12 myotubes which appeared larger with a high number of nuclei per fibre (Fig 2F), indicating that muscle differentiation of C2C12 myoblasts exposed to IGF2R blockade proceeded for those cells that started prematurely to fuse. In line, human primary myoblasts (HSkM) treated with anti-IGF2R were able to form myotubes with long shape and low number of nuclei especially after 6 days of differentiation (Fig 2F). Moreover, HSkM exposed to IGF2R blockade showed increased expression of myogenin and MyHC between days 2 and 4 with fusion capacity similar to C2C12 treated with anti-IGF2R (Fig 2G and H). Together, these data provide evidence indicating the possibility that the timing

of expression of early and late differentiation markers is altered in C2C12 and HSkM myoblasts exposed to IGF2R blockade and that CD20 silencing dampens the effect of IGF2R blockade. We reasoned that this could depend either on a premature differentiation of myoblasts exposed to IGF2R blockade or on the presence of a mixed population of proliferating and differentiating cells.

## IGF2R and CD20 interactions modulate intracellular Ca²⁺ concentrations and activate the CAMKII/calcineurin/NFAT signalling pathway of myoblasts

We thus investigated the mechanisms that could potentially mediate IGF2R blockade-induced myoblast differentiation. Western blotting showed that blockade of IGF2R remarkably increased CAMKII phosphorylation in C2C12 myoblasts but did not affect shCD20-treated C2C12 myoblasts (Fig EV2A and B). Conversely, blockade of IGF2R decreased the expression of calcineurin in C2C12 myoblasts (Fig EV2C and D). Moreover, the nuclear expression of NFAT was increased in store depletion (SD)− and SD+anti-IGF2R-treated

C2C12 myoblasts, and this change was correlated with an increase in the level of MyoD (Fig EV2E and F). These data suggest that SD and blockade of IGF2R in SD myoblasts activate CaMKII and negatively regulate calcineurin whereas the net effect is to induce NFAT to activate nuclear MyoD expression and myoblast differentiation.

To determine whether IGF2R and CD20 interactions may modulate intracellular $Ca^{2+}$ concentrations, we measured Fluo-4-loaded C2C12 and C2C12 shCD20 myoblasts previously treated with or without anti-IGF2R antibodies. Cells treated with anti-FLAG antibodies were assayed as a negative control. The percentage of Fluo-4$^+$ C2C12 cells was significantly higher in cells treated with anti-IGF2R antibodies than in both untreated ($P < 0.05$) and anti-FLAG ($P < 0.005$)-exposed cells (untreated: $20.19\% \pm 8.12\%$; anti-FLAG: $15.3\% \pm 3.10\%$; anti-IGF2R: $36.4\% \pm 11.5\%$) (Fig EV3A), suggesting that myoblasts are responsive to SOCE after IGF2R binding. Additionally, the mean fluorescence intensity (a value directly proportional to the amount of intracellular $Ca^{2+}$) was slightly higher in anti-IGF2R-treated C2C12 myoblasts than in untreated cells (untreated: $28.07 \pm 5.31$; anti-IGF2R: $35.02 \pm 9.14$) ($P < 0.05$) (Fig EV3B), indicating that SOC currents were modestly increased by exposure to anti-IGF2R antibodies. Importantly, the proportion of CD20$^+$ cells in the total number of Fluo-4$^+$ C2C12 cells was quantified, and on average, 32% of cells were double-positive (Fig EV3C). However, Fluo-4$^-$CD20$^+$ cells were 57.4%, indicating that not all of the C2C12 myoblasts that expressed CD20 were responsive to SOCE under these experimental conditions. Together, these data indicate that responsiveness to SOCE can be unlocked and SOC influx enhanced by targeting CD20 in C2C12 myoblasts, supporting the notion that CD20 acts as an SOCE modulator in these cells. We next carried out experiments to investigate how the loss of CD20 affects SOCE in C2C12 myoblasts. Measurement of SOCE showed that a significantly higher percentage of Fluo-4$^+$ cells were observed in the CD20-silenced than in untreated C2C12 myoblasts, indicating that responsiveness to SOCE was consistently enhanced in these cells (untreated: $33.8\% \pm 2.33$; shRNA CD20: $46.8\% \pm 2.11$; $P < 0.05$). Importantly, there was no significant difference in the behaviour of cells infected with shRNA construct that encoded a scrambled sequence and untreated C2C12 myoblasts, confirming the specificity of this phenomenon (shRNA CTR: $18.9\% \pm 9.88$; Fig EV3D). Given that the knockdown efficiency was below 70%, exposure to the anti-CD20 antibody was expected to at least partially recapitulate the effects observed in untreated (not silenced) C2C12 myoblasts. Treatment with anti-IGF2R antibodies also increased the proportion of Fluo-4$^+$ cells in CD20-silenced C2C12 myoblasts (Fig EV3D). In contrast, there was no significant difference in mean fluorescence intensity between untreated and CD20-silenced C2C12 myoblasts regardless of whether they were or were not exposed to anti-IGF2R antibodies, suggesting that the achieved knockdown efficiency was not sufficient to affect SOC influx amplitudes (Fig EV3E).

## IGF2R blockade regulates $Ca^{2+}$ homeostasis of myoblasts via store-operated calcium entry (SOCE) and SERCA1

We next carried out continuous fluorescence recording experiments to address whether targeting CD20 would modulate the magnitude of SOCE in C2C12 myoblasts. Fluo-4-loaded cells were imaged in real time beginning during the intracellular $Ca^{2+}$ SD step and throughout the process during which the external $Ca^{2+}$

concentration was increased so that both the preparative SD and effective ($Ca^{2+}$ re-entry) phases of SOCE were monitored. Compared to untreated cells, the anti-IGF2R-treated myoblasts ($n = 133$) had a significantly lower fluorescence profile throughout the imaging period, indicating that both SD and $Ca^{2+}$ re-entry were impaired (Fig EV3F). The effective phase of SOCE was impaired in CD20-silenced C2C12 myoblasts treated with anti-IGF2R antibodies ($n = 49$), suggesting an interaction between IGF2R and CD20 during the SD step (Fig EV3F). Cells treated with control antibodies (anti-FLAG; $n = 33$) or infected with an shRNA encoding a scrambled sequence ($n = 39$) behaved in a manner almost identical to that observed in untreated cells ($n = 104$ and $n = 100$, respectively) at all time points, confirming the specificity of our observations (Fig EV3H).

To assess whether IGF2R targeting specifically modulated SOCE, cells were assayed in the presence of the SOC influx inhibitor BTP2. When the anti-IGF2R-treated ($n = 40$) and CD20-silenced C2C12 myoblasts ($n = 42$) were compared to cells receiving the SOCE blocker alone ($n = 48$), no difference was observed, suggesting that IGF2R targeting is ineffective if SOCE does not occur (Fig EV3I). We further assessed SD efficiency by adding 10 mM caffeine after the SD protocol. The caffeine-induced intracellular $Ca^{2+}$ peak was almost completely abolished in untreated ($n = 40$) and CD20-silenced cells ($n = 41$), demonstrating that SR stores are effectively emptied. In contrast, a $Ca^{2+}$ peak was clearly observed in cells exposed to anti-IGF2R antibodies ($n = 46$), indicating that SD efficiency was reduced by blockade of IGF2R (Fig EV3G). The impact of CD20 targeting on the cellular response to a $Ca^{2+}$-free environment was also investigated before thapsigargin was added. As expected, in C2C12 myoblasts ($n = 30$), the intracellular $Ca^{2+}$ concentration progressively decreased over time. This decrease was significantly delayed in anti-IGF2R-treated cells ($n = 39$), suggesting a role for CD20 in mediating the cell response to extracellular $Ca^{2+}$ deprivation (Fig EV3J).

Because the plasma membrane channel ORAI1 is the main component of SOCE, we investigated its interaction with CD20 and its expression in SD myoblasts treated with or without anti-IGF2R. As expected, we found that ORAI1 and CD20 co-immunoprecipitated (Fig EV3K and L). Blockade of IGF2R increased CD20 expression, whereas increased CD20 phosphorylation was observed in SD− and SD+anti-IGF2R-treated myoblasts (Fig EV4A and B). Conversely, CD20 phosphorylation was impaired or reduced in SD shCD20 and normal myoblasts after IFG2R blockade (Fig EV4A and B). Moreover, the CD20 phosphorylation induced by blocking IGF2R in myoblasts decreased the interaction between CD20 and ORAI1 in SD myoblasts and mostly blocked this interaction in SD myoblasts treated with anti-IGF2R antibodies (Fig EV3J and K), suggesting a possible interaction between ORAI1 and STIM1 leads to $Ca^{2+}$ release from intracellular calcium stores in the SR. Interestingly, SERCA1 expression was increased in ER lysates of anti-IGF2R-treated SD C2C12 cells compared to the SD+CD20-anti-IGF2R-treated C2C12 myoblasts (Fig EV4C). Since SOCE modulates SERCA activity, we tested ATPase hydrolysis and found that the myogenic differentiation induced by blockade of IGF2R significantly increases SERCA activity compared to shCD20 SD+anti-IGF2R cells ($P < 0.0001$ after the first 2 min and $P < 0.01$ after 3 min; Fig EV4D). The SD and SD+anti-IGF2R showed increased SERCA activity compared to the untreated C2C12 cells ($P < 0.0001$ for the

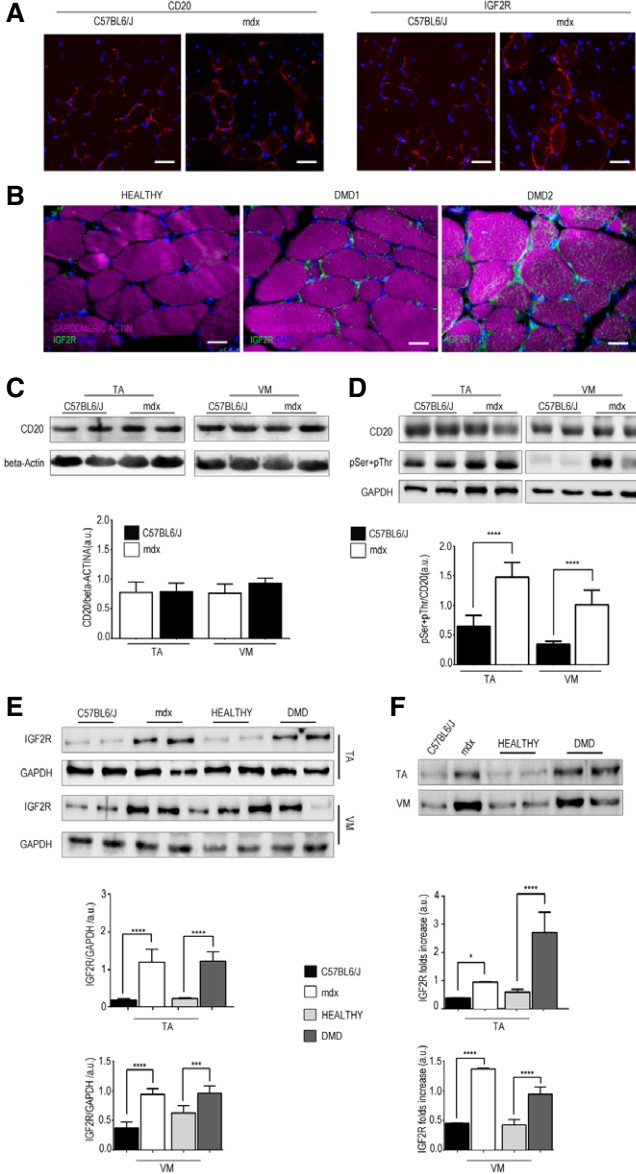

**Figure 3. IGF2R is over-expressed in dystrophic muscle.**

A  Immunofluorescence analysis of CD20 and IGF2R expression (in red) in the TA muscles of three-month-old C57Bl6/J and *mdx* mice. Scale bars = 75 μm.

B  IGF2R (green) and sarcomeric actin (magenta) expression in the VM muscles of two DMD patients. Scale bars = 25 μm.

C, D  Representative WB analysis of CD20 and β-actin (C) and pSer + pThr and GAPDH (D) expression in the TA and VM muscles of C57Bl6/J and *mdx* mice (*n* = 10 mice per group). Densitometry analysis of WB data is expressed as the CD20/β-actin ratio and pSer + pThr/CD20 ratio in arbitrary units in the lower panels. One-way ANOVA. ****P < 0.0001. All values are expressed as the mean ± SEM.

E  Representative WB analysis of IGF2R and GAPDH expression in total protein lysates of the human (healthy and DMD) and murine (C57Bl6/J and *mdx*) TA and VM muscles (*n* = 2 per healthy and DMD; *n* = 10 mice per group); quantifications are expressed as the IGF2R/GAPDH ratio in arbitrary units. One-way ANOVA. ***P < 0.001; ****P < 0.0001. All values are expressed as the mean ± SEM.

F  Representative WB analysis of IGF2R expression in sarcolemma protein lysates of the TA and VM muscles of healthy, DMD, C57Bl6/J and *mdx* (*n* = 2 per healthy and DMD; *n* = 10 mice per group). Densitometry analysis of IGF2R expression, results are shown as a fold increase, as indicated in the lower panel in arbitrary units. One-way ANOVA. *P < 0.05; ****P < 0.0001. All values are expressed as the mean ± SEM.

Source data are available online for this figure.

protein is localized. For this reason, when we evaluated IGF2R protein expression levels in skeletal muscle tissues, we performed WB experiments using both total protein extracts and isolated sarcolemma. These expression patterns showed significant increase of IGF2R in both total and sarcolemmal extracts of the *Tibialis Anterior* (TA) and *Vastus Medialis* (VM) muscle tissues of mdx mice and DMD patients than those observed in healthy controls (Fig 3 E and F). Moreover, the pattern of low IGF2R expression in healthy muscles (Fig 3E and F) may reflect that IGF2 is mainly expressed during the development and dramatically reduced after birth and in adult tissues (de Pagter-Holthuizen *et al*, 1987). No bands were obtained when GAPDH blotting was performed using isolated sarcolemma, confirming the functionality of the isolation protocol.

### IGF2R blockade ameliorates muscle function in *mdx* mice

To study the effects of IGF2R blockade on muscular dystrophy, we intravenously administered anti-IGF2R antibodies at low (10 μg per mouse) and high (100 μg per mouse) dosages to 3-month-old *mdx* mice for 4 and 9 weeks. This genetically dystrophic mouse model exhibits dystrophic muscle features and skeletal muscle vascular regression (Loufrani *et al*, 2001, 2002, 2004; Williams & Allen, 2007). Morphometric analysis of the TA and VM muscles of anti-IGF2R-treated *mdx* mice showed that the levels of typical fibrotic infiltrate and centrally nucleated fibres were lower than in controls (Fig EV5). *Mdx* mice were characterized by high variability in myofibre size (Fig EV5). The cross-sectional areas (CSAs) of the myofibres were significantly lower in the muscles of *mdx* mice treated with a low dose of anti-IGF2R for 4 weeks (TA: $2,251 \pm 33$ μm$^2$, $n = 10$; VM: $1,882 \pm 41$ μm$^2$, $n = 10$) than in the muscles of untreated age-matched mice (TA: $4,476 \pm 94$ μm$^2$, $n = 10$; VM: $1,383 \pm 73$ μm$^2$, $n = 10$) ($P < 0.0001$, one-way ANOVA analysis of variance with Bonferroni correction). Although the difference was not significant, a similar reduction in the CSA

first 2 min, $P < 0.001$ after 3 min, $P < 0.01$ after 4 min, $P < 0.05$ after 5 min, $P < 0.0001$ for the first 4 min, $P < 0.01$ after 5 min and $P < 0.05$ after 6 min, respectively; Fig EV4E).

### IGF2R is highly expressed in dystrophic muscles

The skeletal muscles of *mdx* mice exhibited similar CD20 expression levels but increased IGF2R expression compared to that observed in C57Bl6/J mice (Fig 3A and C). Similarly, IGF2R expression was increased in human dystrophic muscles obtained from two DMD patients compared to healthy human muscles (Fig 3B and E). We found that the level of CD20 phosphorylation was higher in *mdx* muscle and that this change was related to an alteration in IGF2R expression (Fig 3D and E). Due to its involvement in the transport of lysosomal enzymes and IGF lysosomal degradation, IGF2R is continuously recruited to the intracellular space and then recycled back to the cell membrane, in which only 10–20% of the total IGF2R

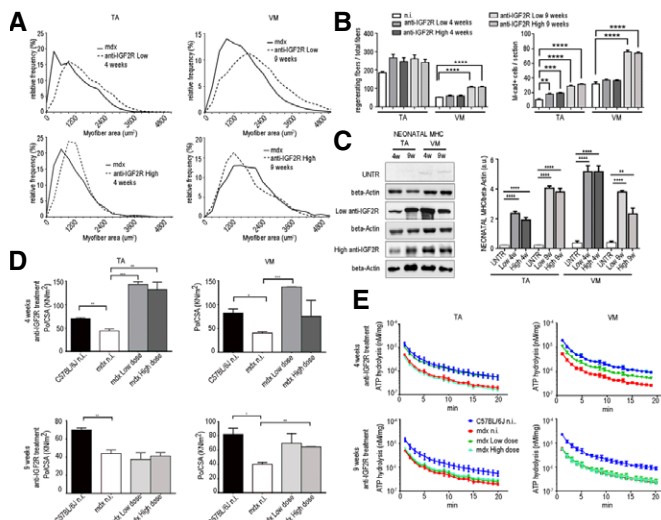

**Figure 4. Blockade of IGF2R ameliorates muscular dystrophy.**

A  Quantification of the relative frequency of myofibre cross-sectional areas (CSAs) expressed as a frequency distribution in the TA and VM muscles of untreated *mdx* mice or *mdx* mice treated with a low or high dose of anti-IGF2R for 4 and 9 weeks.

B  Quantification of regenerating myofibres as a proportion of centrally nucleated fibres in the total number of fibres after anti-IGF2R administration. The quantification of regenerating and M-cadherin-positive satellite cells per section is shown. One-way ANOVA test, **$P < 0.01$; ***$P < 0.001$; ****$P < 0.0001$. Each experiment was performed in triplicate wells, and all values are expressed as the mean $\pm$ SEM.

C  Representative WB analysis of neonatal myosin heavy-chain (neonatal MHC) expression and quantification expressed as the ratio of neonatal MHC/β-actin in untreated *mdx* mice and *mdx* mice treated with low and high doses of anti-IGF2R for 4 and 9 weeks. One-way ANOVA. **$P < 0.01$; ****$P < 0.0001$ ($n = 10$ mice per group). All values are expressed as the mean $\pm$ SEM.

D, E  Specific force (maximal tetanic force normalized to cross-sectional area, Po/CSA) from TA and VM muscles (D) and SERCA activity quantified as ATPase hydrolysis activity vs. min (E) in C57Bl6/J mice, untreated *mdx* mice and *mdx* mice treated with low and high doses of anti-IGF2R for 4 and 9 weeks. One-way ANOVA. *$P < 0.05$; **$P < 0.01$; ***$P < 0.001$ ($n = 10$ mice per group). All values are expressed as the mean $\pm$ SEM.

Source data are available online for this figure.

was observed in the muscles of *mdx* mice treated with a high dose of anti-IGF2R for 4 weeks (TA: $2,847 \pm 63.5$ μm$^2$, $n = 10$; VM: $1,627 \pm 42$ μm$^2$, $n = 10$) compared to the CSAs observed in the muscles of untreated *mdx* mice (TA: $4,110 \pm 80.3$ μm$^2$, $n = 10$; VM: $1,668 \pm 19$ μm$^2$, $n = 10$; Fig 4A). Moreover, the coefficient of variation of myofibre area was significantly higher for the muscles of the anti-IGF2R-treated low dosage *mdx* mice (TA: $73.22 \pm 6.17\%$ for 4 weeks and $59 \pm 5.93\%$ for 9 weeks; VM: $74 \pm 4.81\%$ for 4 weeks and $66 \pm 3.38\%$ for 9 weeks) than for those of untreated *mdx* mice (TA: $50.48 \pm 5.4\%$; VM: $47.37 \pm 7.5\%$) ($P < 0.0001$, F-test of variance). However, the areas of the muscle fibres were significantly higher in the high-dose anti-IGF2R-treated *mdx* mice (TA: $4,609 \pm 57$ μm$^2$ for 4 weeks and $5,188 \pm 72$ μm$^2$ for 9 weeks; VM: $3,353 \pm 41$ μm$^2$ for 4 weeks and $4,704 \pm 73$ μm$^2$ for 9 weeks) than in the untreated mice (TA: $2,006 \pm 55$ μm$^2$; VM: $2,705 \pm 41$ μm$^2$; Fig 4A). CSA values were also significantly reduced in the TA ($1,709 \pm 20$ μm$^2$, $n = 10$) and VM

($2,270 \pm 23$ μm$^2$) muscles of *mdx* mice treated with a high dose of IGF2R for 4 weeks than in the TA ($1,885 \pm 20.5$ μm$^2$) and VM ($3,002 \pm 33$ μm$^2$) muscles of untreated *mdx* mice. A similar trend was observed following 9 weeks of treatment (TA: untreated $2,916 \pm 85$ μm$^2$ vs. treated $2,096 \pm 53$ μm$^2$; VM: untreated $2,460 \pm 60$ μm$^2$ vs. treated $2,773 \pm 40$ μm$^2$) ($P < 0.001$, one-way ANOVA with Bonferroni correction; Fig 4A). Changes in the CSAs of the muscles of *mdx* mice suggested that a muscle remodelling process was induced by anti-IGF2R antibodies. Since the IGF1 and IGF2 pathways are closely associated with muscle regeneration, we evaluated the numbers of regenerating neonatal MyHC-positive fibres and M-cadherin-positive satellite cells in untreated and IGF2R-treated *mdx* mice (Fig 4B). We observed that the percentages of regenerating muscle fibres and the level of neonatal MyHC expression were significantly higher in the TA and VM of IGF2R-treated *mdx* mice than in untreated *mdx* mice ($P < 0.001$, one-way ANOVA with Bonferroni correction) (Fig 4B and C). Moreover, the number of M-cadherin-positive satellite cells was similarly increased in the low-dose IGF2R-treated (TA treated for 4 weeks at 20–25/section and for 9 weeks at 30–40/section; VM treated for 4 weeks at 30–40/section and for 9 weeks at 70–80/section) and high-dose IGF2R-treated (TA treated for 4 weeks at 30–41/section and for 9 weeks at 47–51/section; VM treated for 4 weeks at 33–46/section and for 9 weeks at 78–85/section) *mdx* mice (Fig 4B). To confirm these findings, we verified whether muscle function was recovered in the IGF2R-treated mdx mice and evaluated the maximum tetanic force (Po) normalized for the cross-sectional areas (CSA). We found that specific force was ameliorated in the anti-IGF2R-treated *mdx* mice ($P < 0.05$, $P < 0.001$, two-way ANOVA with Bonferroni correction; Fig 4D). Interestingly, the specific force generated by TA muscles of mice treated with both low and high doses of anti-IGF2R and VM muscles of mice treated with low dose of anti-IGF2R for 4 weeks was higher than TA and VM muscles of control C57Bl6/J (Fig 4D). The increased specific force and the reduction of CSA of these anti-IGF2R cohorts indicate similar values of Po between anti-IGF2R-treated *mdx* and C57Bl6/J mice. Since skeletal muscle performance is modulated by SERCA activity and this may explain the prevalence of myofibres expressing regenerating neonatal MyHC in IGF2R-treated *mdx* mice, we compared ATP hydrolysis between IGF2R-treated and untreated *mdx* mice. ATP hydrolysis was significantly increased in the TA ($P < 0.0001$ for the first 3 min and $P < 0.01$ after 4 and 5 min for low dosages after 4 weeks; $P < 0.05$ for the first minute for high dosages after 9 weeks) and VM ($P < 0.0001$ for the first 4 min, $P < 0.001$ after 5 min, $P < 0.01$ after 6 min and $P < 0.05$ after 7 min for high dosages after 4 weeks, two-way ANOVA with Bonferroni correction) muscles of 4- and 9-week low-dose anti-IGF2R-treated mice than in untreated *mdx* mice (Fig 4E). The level of active calcineurin A 48 kDa protein expression was also significantly higher in the TA and VM ($P < 0.05$, $P < 0.001$, two-way ANOVA with Bonferroni correction), whereas the level of inactive calcineurin A 60 kDa was lower in both the TA and VM muscles of anti-IGF2R-treated than in untreated *mdx* mice (Fig 5A). In contrast, CAMKII phosphorylation was significantly decreased in both the TA and VM muscles of anti-IGF2R-treated *mdx* mice (Fig 5B). Thus, we sought to investigate the effect of this cascade of signals on the PKC-α downstream signalling pathway. In response to anti-IGF2R, there was no statistically significant effect on PKC-α expression or phosphorylation in the TA and VM

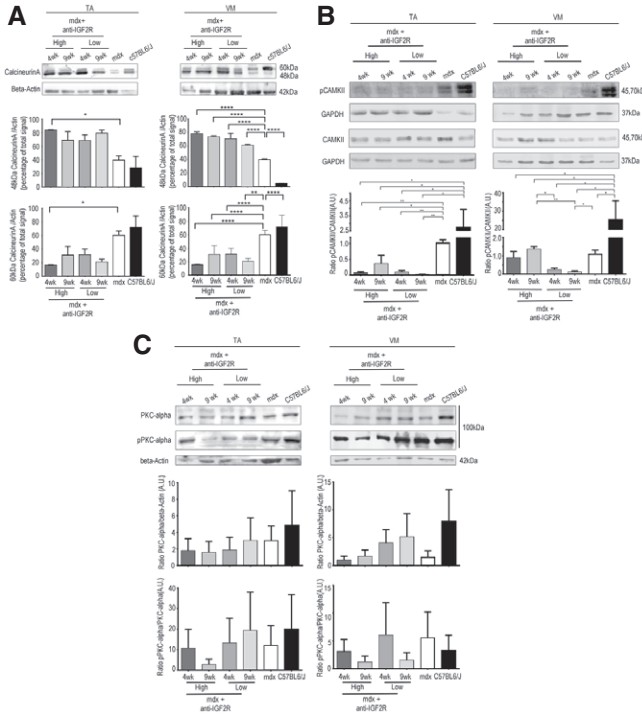

**Figure 5. Calcineurin signalling activation in muscles of *mdx* mice treated with anti-IGF2R.**

A  Representative WB analysis of active 48 kDa and inactive 60 kDa calcineurin A and β-actin expression in the TA and VM muscles in untreated C57BL mice, untreated *mdx* mice and *mdx* mice treated with low and high doses of anti-IGF2R for 4 and 9 weeks. Densitometry analysis of data expressed as ratios of 48 kDa and 60 kDa calcineurin A/β-actin in arbitrary units. Two-way ANOVA. *P < 0.05; **P < 0.01; ****P < 0.0001 (n = 10 mice per group). All values are expressed as the mean ± SEM.

B  Representative WB analysis of CAMKII, pCAMKII and GAPDH expression quantified and shown as CAMKII/GAPDH and pCAMKII/CAMKII ratios in arbitrary units. Two-way ANOVA. *P < 0.05; **P < 0.01 (n = 10 mice per group). All values are expressed as the mean ± SEM.

C  Representative WB analysis of PKCα, pPKCα and β-actin expression levels quantified and expressed as the PKCα/β-actin and pPKCα/PKCα ratios in arbitrary units (n = 10 mice per group). All values are expressed as the mean ± SEM.

Source data are available online for this figure.

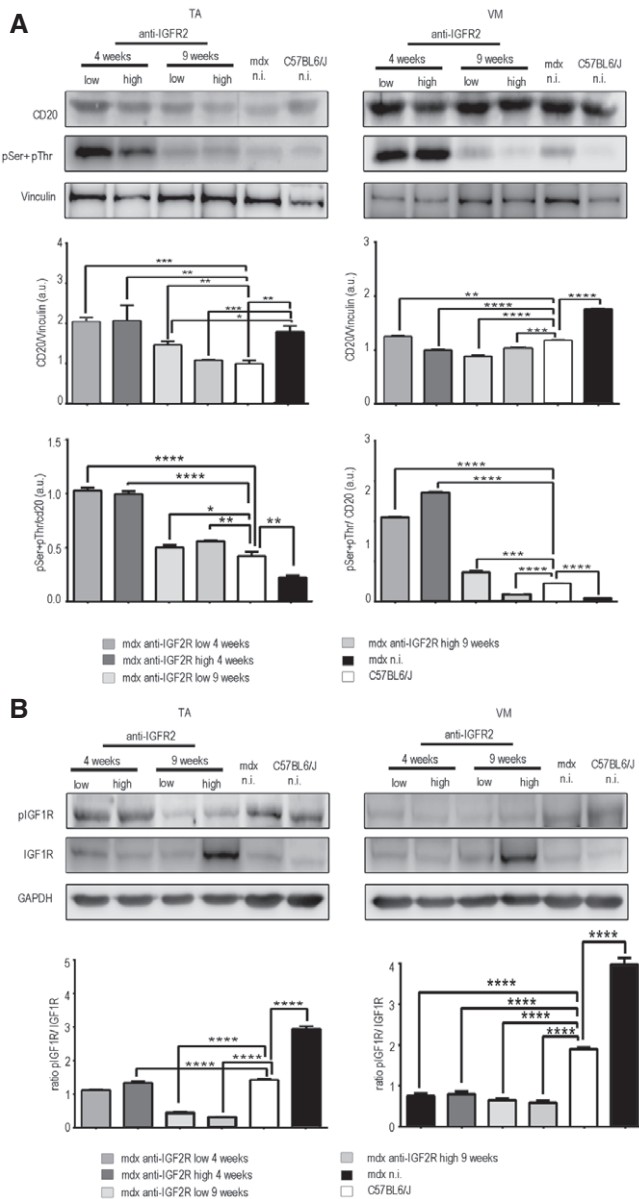

**Figure 6. IGF2R blockade results in CD20 and IGF1Rβ phosphorylation in mdx mice.**

A  WB analysis performed in TA and VM muscles of *mdx* mice treated with anti-IGF2R using antibodies against CD20, pSer + pThr and vinculin. Densitometry analysis of data expressed as the pSer + pThr/CD20 and CD20/vinculin ratio in arbitrary units. One-way ANOVA *P < 0.05; **P < 0.01; ***P < 0.001; ****P < 0.0001 for the comparison with untreated mice (n = 10 mice per group). All values are expressed as the mean ± SEM.

B  Representative WB analysis performed on TA and VM muscles of *mdx* mice treated with anti-IGF2R using anti-IGF1R, anti-phospho-IGF1R and GAPDH. Densitometry analysis of data expressed as the pIGF1R/IGF1R ratio normalized to GAPDH content in arbitrary units. One-way ANOVA. ****P < 0.0001 for the comparison with untreated mice (n = 10 mice per group). All values are expressed as the mean ± SEM.

Source data are available online for this figure.

muscles (Fig 5C). Thus, we next evaluated CD20 expression and phosphorylation as an upstream marker of the effect of anti-IGF2R treatment on signalling. After 4 and 9 weeks, treatment with a low dose of anti-IGF2R significantly increased CD20 expression in the TA muscle (*P* < 0.001, two-way ANOVA with Bonferroni correction) (Fig 6A). Similarly, CD20 phosphorylation was significantly higher in TA and VM muscles of treated vs. untreated mice (*P* < 0.05, two-way ANOVA with Bonferroni correction) (Fig 6A). Interestingly, IGF1R phosphorylation was increased in the TA muscles of mice treated with a low dose of anti-IGF2R and in the VM muscles of mice treated with both low and high doses of anti-IGF2R (Fig 6B), suggesting that IGF1 and IGF2 bioavailability was increased, allowing them to interact with IGF1R, following IGF2R blockade. To test this possibility, we performed experiments to evaluate IGF1R phosphorylation after treatment with anti-IGF2R in 3T3 mouse fibroblasts (Fig EV1C–E). IGF1R phosphorylation was

increased after IGF2 was added to the cell cultures (Fig EV1C). When cells were co-stimulated with IGF2 and anti-IGF2R, the level of IGF1R tyrosine phosphorylation was higher in than the effect

observed following treatment with IGF2 alone, confirming the notion that IGF bioavailability is associated with the activation of IGF1R (Fig EV1D).

## Anti-IGF2R treatment ameliorates vascular rarefaction in the muscles of *mdx* mice

Since the IGF2 pathway affects vascular architecture, we examined the effect of anti-IGF2R on blood vessels in the skeletal muscles of *mdx* (*n* = 10 each group) mice. To analyse the effect of anti-IGF2R on the vessel structures of dystrophic skeletal muscles, we performed vessel corrosion casting electron microscopy to examine capillary architecture and mural cell distribution. Using this approach, it is possible to identify some of artefacts caused by the pressure of injection, treatment with dehydration and other manipulations involved in SEM analysis. One of the most frequently observed types of artefacts is caused by excessive injection pressure and manifests as vascular leakage of the resin that appears in the form of roundish or sheet-like conglomerates extending around the vessels. Although in normal tissues, this event is mostly caused by the technique itself, in pathological tissues, it may also be caused by impaired endothelial cell junctions, modifications of the vascular wall architecture or a rupture in the endothelial cell lining. In untreated *mdx* mice, muscle capillaries had no constant direction, displayed many enlargements and constrictions and were frequently lacking coverage by pericytes, indicating an immature vessel architecture (Fig 7A). In contrast, muscle capillaries were linearized in anti-IGF2R-treated *mdx* mice (Fig 7A). In particular, in treated VM muscles, the number of vascular leakages was slightly lower, the vasculature had a less chaotic and complex architecture, and vessel coverage by pericytes was improved (Fig 7A). Interestingly, we found that the number of pericyte-like cells co-expressing NG2 and CD146 was higher in the treated animals than in the untreated *mdx* mice (Fig 7B). NG2$^+$ cells significantly increased in anti-IGFR2-treated *mdx* mice after 9 weeks of treatment with a low dose (one-way ANOVA, $P < 0.001$) and after 4 and 9 weeks of treatment with a high dose (one-way ANOVA, $P < 0.0001$) (Fig 7C). Similarly, CD146$^+$ cells significantly increase in anti-IGFR2-treated *mdx* mice after 9 weeks of treatment with a low dose and after 4 and 9 weeks of treatment with a high dose (one-way ANOVA, $P < 0.0001$; Fig 7C). We also analysed vessel leakage in detail by immunofluorescence and found that the VM muscles of anti-IGF2R-treated *mdx* mice exhibited highly uniform alignments of VE-cadherin and CD31-expressing cells compared to untreated *mdx* mice (Fig 7E). Quantification of CD146 and NG2 expression showed that CD146 expression in TA skeletal muscles was significantly increased in the low-dose anti-IGF2R-treated mice after 4 and 9 weeks (one-way ANOVA, $P < 0.0001$ and $P < 0.001$) and in high-dose anti-IGF2R-treated mice after 4 weeks (one-way ANOVA, $P < 0.01$; Fig 7D). However, NG2 expression was significantly higher in low-dose anti-IGF2R-treated mice after 9 weeks (one-way ANOVA, $P < 0.001$) and in high-dose anti-IGF2R-treated mice after 4 and 9 weeks (one-way ANOVA, $P < 0.05$). Interestingly, the expression of CD146 and NG2 in VM muscles showed significant increase in all conditions tested (Fig 7D). Together, these observations indicate that IGF2R blockade in *mdx* mice promoted vascular remodelling and thereby led to muscle capillary linearization and leakage reduction by increasing the number of pericytes.

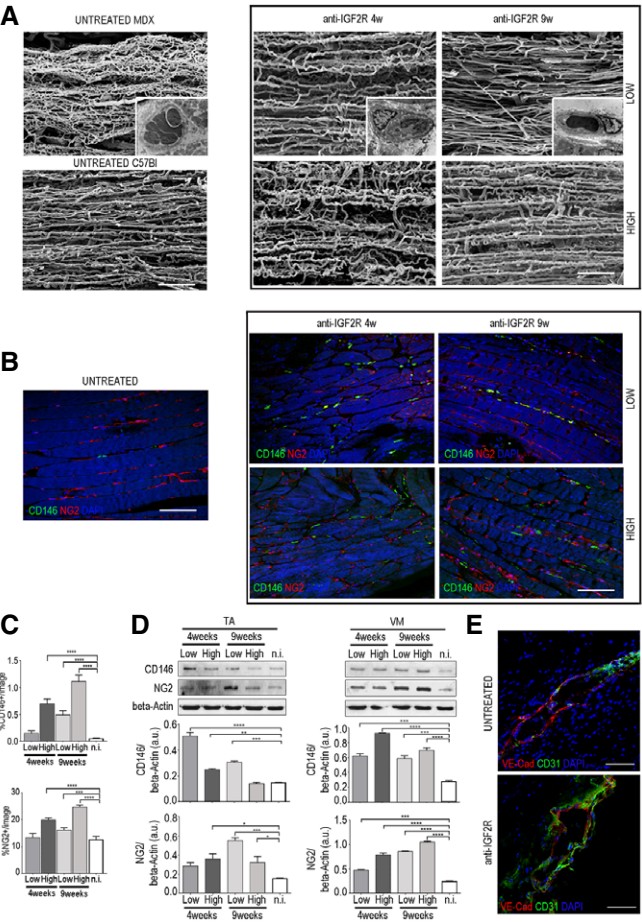

**Figure 7. IGF2R blockade induces normalization of capillary architecture in *mdx* mice.**

A   Corrosion casting results showing the linearization of capillaries in IGF2R-treated VM *mdx* muscles. Pericytes around capillaries are shown in the insert box. Scale bars = 100 μm.

B, C   Representative immunofluorescence results (B) and quantification (C) of longitudinal sections of VM *mdx* muscles for the CD146- and NG2-positive capillaries are shown. Scale bars = 75 μm. One-way ANOVA. ***$P < 0.001$; ****$P < 0.0001$ for the comparison with untreated mice (*n* = 10 mice per group). All values are expressed as the mean ± SEM.

D   Representative WB showing CD146 and NG2 expression in total protein lysates of TA and VM muscles of *mdx* mice treated with anti-IGF2R. Densitometry analysis of data expressed as the ratio of the indicated antibody/β-actin in arbitrary units. One-way ANOVA. *$P < 0.05$; **$P < 0.01$; ***$P < 0.001$; ****$P < 0.0001$ for the comparison with untreated mice (*n* = 10 mice per group). All values are expressed as the mean ± SEM.

E   Representative VE-cadherin (red) and CD31 (green) expression in small vessels of VM muscles from untreated *mdx* and *mdx* treated with low dose of anti-IGF2R for 4 weeks. Scale bars = 25 μm.

Source data are available online for this figure.

## Discussion

In this study, we explored the role(s) of IGF2R in modulating the activity of endogenous IGFs in the context of dystrophic muscle. We show that IGF2R is over-expressed in the skeletal muscles of *mdx* mice, an animal model of DMD. Intravenous administration of an

anti-IGF2R-neutralizing antibody consistently increased muscle regeneration and reduced fibrosis and thereby significantly ameliorated the dystrophic muscle phenotype. These data strongly argue that IGF2R inhibition positively affected the regenerative potential of dystrophic muscle tissues. This conclusion makes physiological sense in the light of what is known about the role of IGF2R in the IGF system, in which it is responsible for IGF internalization and lysosomal degradation. For example, among the variety of hormones and growth factors known to be involved in myogenesis regulation, IGFs (including IGF1 and IGF2) exert very critical roles (Musaro *et al*, 2001, 2004). Mice deficient in IGF ligands exhibit muscle hypoplasia and die shortly after birth because they lack the muscle mass required to inflate their lungs (Liu *et al*, 1993; Powell-Braxton *et al*, 1993), and transgenic mice that over-express IGF1 in muscles have larger muscle fibres and enhanced muscle strength during ageing (Coleman *et al*, 1995; Barton-Davis *et al*, 1998; Barton *et al*, 2002; Kaspar *et al*, 2003). In cultured muscle cells, IGF2 levels increase dramatically during myogenesis, IGF2 antisense oligonucleotides abolished this differentiation process (Florini *et al*, 1991, 1996), and IGF2 over-expression accelerated myoblast differentiation (Stewart *et al*, 1996). Moreover, Rotwein and colleagues reported that in cultured muscle cells, secreted IGF2 stimulated IGF1R, PI3K and Akt to induce the expression of the cyclin-dependent kinase inhibitor p21 and myogenin (Lawlor & Rotwein, 2000; Wilson & Rotwein, 2007). The muscle impairment observed in DMD patients is characterized by extremely low regenerative capacity, fibrosis, fat deposition, inflammatory infiltrates and fibre hypotrophy. In recent decades, a number of different approaches have been investigated to treat DMD. Unfortunately, most therapeutic strategies have been palliative rather than curative. In this context, the benefits obtained after IGF2R blockade prompted us to explore a new potential therapeutic approach. In this study, we provide the first data showing that CD20 binds to the conserved domain 11 of IGF2R. Importantly, domain 11 was the first of the IGF2R extracellular regions to be characterized by X-ray crystallography, and it contains the putative IGF-binding site (Schmidt *et al*, 1995; Brown *et al*, 2002). Our experimental findings provide evidence indicating that IGF2R specifically interacts with CD20 and blockade of IGF2R induces CD20 phosphorylation promoting intracellular $Ca^{2+}$ modifications.

Muscular dystrophies (MD) are often associated with $Ca^{2+}$ dyshomeostasis. Intracellular $Ca^{2+}$ ions critically regulate contraction and force production of muscle fibres by acting as the primary regulator of the sarcomeric contractile machinery and as a second messenger in the signal transduction pathways that control muscle growth, metabolism and pathological remodelling (Berchtold *et al*, 2001; Bassel-Duby & Olson, 2006). During exercise, $Ca^{2+}$ is cycled between the cytosol and the sarcoplasmic reticulum (SR) through a system by which the $Ca^{2+}$ pool in the SR is restored by uptake of extracellular $Ca^{2+}$ via a mechanism called store-operated $Ca^{2+}$ entry (SOCE). Among the proteins known to be involved in the activation and regulation of SOCE, the two most prominent are STIM1 and ORAI1. STIM1 spans the SR membrane and, through its SR-hand domain located in the lumen of the SR cistern, senses the $Ca^{2+}$ concentration in cellular stores. Depletion of ER calcium results in the horizontal movement of STIM1 in the SR membrane, causing it to cluster and interact with the plasma membrane channel-forming protein ORAI1. This eventually results in $Ca^{2+}$ entry into the cell.

Replenishment of $Ca^{2+}$ stores breaks these molecular interactions and stops $Ca^{2+}$ influx (Smyth *et al*, 2010; Gudlur *et al*, 2013). Interestingly, the CD20 phosphorylation induced by IGF2R blockade in myoblasts decreased the interaction between CD20 and ORAI1 in store-depleted myoblasts, and this effect was increased in store-depleted myoblasts treated with anti-IGF2R. These data corroborate the hypothesis (Ju *et al*, 2003; Parolini *et al*, 2012) that CD20 interacts with ORAI1 in the muscle plasma membrane and that its phosphorylation promotes the interaction between ORAI1 and STIM1, which mediates $Ca^{2+}$ release from intracellular calcium stores in the SR.

Reuptake of $Ca^{2+}$ ions in the SR during excitation–contraction (EC) coupling is regulated by the ATP-dependent sarcoplasmic/endoplasmic reticulum calcium ATPase pump (SERCA1; Rossi & Dirksen, 2006). Interestingly, SERCA activity has been reported to be reduced in dystrophic muscle (Kargacin & Kargacin, 1996; Divet & Huchet-Cadiou, 2002; Divet *et al*, 2005), and this effect is likely responsible for some characteristics of defective $Ca^{2+}$ handling observed in MD because it leads to higher cytoplasmic levels of $Ca^{2+}$ and increases cellular necrosis via calpain activation and increased mitochondrial permeability (Odermatt *et al*, 1996; Dorn & Molkentin, 2004; Periasamy & Kalyanasundaram, 2007; Goonasekera *et al*, 2011). Blockade of IGF2R activated SERCA1 and enhanced SR $Ca^{2+}$ uptake, promoting premature differentiation of myoblasts and correcting $Ca^{2+}$ overload; these effects likely induced muscle regeneration and vessel development in dystrophic *mdx* mice. CaMKII can activate the reuptake of $Ca^{2+}$ ions in the SR regulating the SERCA1 (Damiani *et al*, 2000; Sacchetto *et al*, 2000). The results of IGF2R blockade of myoblasts suggest that calcineurin inhibits CaMKII-mediated phosphorylation, and the inhibition of calcineurin increases phospho-CaMKII, which results in the stimulation of CaMKII-dependent cellular actions. Otherwise, the IGF2R blockade of *mdx* mice leads to negative regulation of CaMKII and activation of calcineurin. The calcineurin is involved in the control of skeletal myofibre specialization and can transform type II fibres into type I fibres in a dose-dependent manner, and CaMK acts synergistically with calcineurin to activate slow and oxidative fibre-specific gene expression in cultured myoblasts (Wu *et al*, 2000). Previous report suggested that CaMKII-mediated processes were abolished by calcineurin in skeletal muscle (Wu *et al*, 2002). As for $Ca^{2+}$ release from intracellular stores, calcineurin is directly involved in its regulation and calcineurin inhibitors lead to a higher probability of the ryanodine receptor (RyR)/$Ca^{2+}$-release channels being open (Bandyopadhyay *et al*, 2000; Bultynck *et al*, 2003).

Moreover, calcineurin dephosphorylates nuclear factor of activated T cells (NFAT) hereby regulating its nuclear localization and facilitates the increased expression of genes involved in myogenic programme.

Our results further support the idea that blockade of IGF2R increases intracellular $Ca^{2+}$ of myoblasts activating cytoplasmic signalling cascades with opposite effects on calcineurin activity, and the net effect is NFAT dephosphorylation and translocation into the nucleus. Finally, mechanical analysis of intact muscles revealed that muscle force was strongly increased in anti-IGF2R-treated *mdx* mice and these data correlated to increased SERCA1 activity. It is reasonable to conclude from these findings that the increased muscle force observed in these mice represents the result of a synergy among all

the benefits promoted by anti-IGF2R treatment in dystrophic muscle tissue.

Anti-IGF2R binding to domain 11 of IGF2R activates IGF2R-Gαi2 interactions and prevents it from interaction with IGF2. This latter leads to a decrease in the degradation and a consequential increase in the bioavailability of IGF2 for IGF1R interactions with consequent IGF1R phosphorylation, which recruits the PI3-K/Akt/mTOR signalling axis to regulate the expression of skeletal muscle-specific genes, that are associated with myogenic differentiation. The IGF1-induced pathway also increases $Ca^{2+}$ influx via SOCE activation (Berridge, 1995; Parekh & Penner, 1997; Barritt, 1999). Interestingly, PI3-K regulates the CD20 phosphorylation (Ju *et al*, 2003; Balaji *et al*, 2018). Thus, blockade of IGF2R reestablished the correct oscillating pattern of $Ca^{2+}$ ion levels in the microenvironment of myofibrils and protects from MD acting on different mechanisms of dystrophic muscles.

In line with these observations, we found that *in vivo* blockade of IGF2R resulted in a significant remodelling of the characteristically disorganized (i.e. enlarged or constricted) or immature capillaries and microvessels that surround muscle fibres in untreated *mdx* mice. This effect can be explained, at least in part, by the increase in the number of pericyte-like cells with a perivascular localization observed in anti-IGF2R-treated dystrophic muscles. It is important to note that vascular normalization represents a substantial improvement in the context of degenerating dystrophic muscle as this condition can have a serious influence on inflammatory cell migration, nutrient supply and tissue oxygenation; hence, normalization plays a central role in improving the observed dystrophic muscle phenotypes.

In summary, we provide the first evidence demonstrating that IGF2R is over-expressed in DMD and *mdx* muscles and that systemic administration of an anti-IGF2R-neutralizing antibody resulted in the recovery of the dystrophic muscle phenotype, ameliorated vascular architecture defects and improved muscle force. Importantly, we provide evidence of physical and functional interactions between IGF2R and CD20. Perturbing this interaction with anti-IGF2R increased IGF bioavailability to IGF1R and reduced the intracellular $Ca^{2+}$ concentration in dystrophic muscle cells, eventually resulting in an extremely significant amelioration of dystrophic muscle histology and vasculature defects and force performance. We are firmly convinced that increasing our understanding of the contribution of aberrant IGF2R expression to the pathophysiology of DMD will ultimately lead to the development of novel therapeutic approaches. Even if side effects are not completely avoidable, blockade of IGF2R in DMD patients could represent an encouraging starting point for the development of new biological therapies for DMD.

# Materials and Methods

## Ethics statement

All procedures involving living animals were performed in accordance with Italian law (D.L.vo 116/92 and subsequent additions), which conforms to the European Union guidelines. The use of animals in this study was authorized by the National Ministry of Health (protocol number 10/13–2014/2015). Details on animal

welfare and steps taken to ameliorate suffering are included below. Muscle biopsies from two DMD patients (10 and 11 years old) and two healthy subjects (19 and 22 years old) were obtained after informed consent of each patient (or patient's parents) was given to donate the biopsy to the Telethon Biobank (GTB12001) for research purposes. The research procedures described were approved by the ethics committee of the University of Milan (CR937-G). This study was performed in accordance with International Conference on Harmonisation of Good Clinical Practice guidelines, the Declaration of Helsinki (2008) and the European Directive 2001/20/EC.

## Animal models and *in vivo* experiments

Animals were obtained from Charles River Laboratories International, Inc. (Calco, Italy). Normal (*C57BL6/J*) and dystrophic (*mdx C57BL6/10ScSn-DMDmdx/J*) male mice were used throughout this study. We decided to use only males in order to eliminate the sex variable from the randomization procedure. All mice were fed *ad libitum* and allowed continuous access to tap water. Cage population was limited to maximum four animals each to ensure the health and welfare of animals; equipments and facilities were joined into the cages to ameliorate environmental conditions. Animals were deeply anaesthetized with 2% avertin (0.015 ml/kg) prior to sacrifice by cervical dislocation, and all efforts were made to minimize suffering. Three-month-old mice weighing 20 g were systemically injected into the tail vein with 10 μg (low dose) or 100 μg (high dose) of Polyclonal Goat IgG Anti-IGF2R (AF2447, R&D Systems, Minneapolis, MN) (Chen *et al*, 2011; Lee *et al*, 2015) and treated for 4 or 9 weeks. In the control group, the same number of animals received either no treatment or an injection of a sodium chloride solution (0.9%). Dynamic light scatter and nanotracking analysis (NTA) was performed on anti-IGF2R formulation as a precautionary measure to exclude the presence of aggregates triggering immune response (Appendix Fig S1A and B). To minimize selection bias, blinding was maintained during and after the intervention so that people that administered the treatment to the animals and take care of them afterwards were unaware of the treatments. To maintain the lowest possibility to introduce differences in the characteristics of animals allocated to treatment groups, the responsible of the experiments did not reveal the allocation sequence to the people that conducted the experiments until the data were analysed. In order to rule out the possibility of misleading immunoreactivity against goat polyclonal anti-IGF2R, blood samples of untreated and treated mdx mice were collected and stored for ELISA detection of circulating anti-goat antibodies (Appendix Fig S1C). Recognition of light- and heavy-chain goat immunoglobulins of anti-IGF2R were confirmed in muscles of mdx after 4 and 9 weeks of treatment (Appendix Fig S1D). All treated animals did not display any obvious clinical signs of immune reaction against goat anti-IGF2R treatment.

## Cell line treatments and myogenic differentiation

The C2C12 mouse myoblast cell line was commercially obtained from the ATCC (ATCC® CRL1772™). Cells were tested 1 day before and during the experiments to exclude mycoplasma contamination. The culture medium (DMEM with 10% FBS; Thermo Fisher

Scientific) was supplemented with 1 nM or 10 nM recombinant IGF1 (Tebu-bio, France) or 10 ng/ml or 100 ng/ml recombinant IGF2 (Sigma-Aldrich, Germany), and the cultures were then incubated for 2 h or overnight (ON). Anti-CD20 (sc-7736 I20, Santa Cruz Biotechnology, USA), anti-FLAG (F-7425 Sigma-Aldrich) or anti-IGF2R (MAB2447, R&D Systems, USA) antibodies were added at 4 µg/ml to the C2C12 culture medium, and the cultures were incubated for 24 h. Human skeletal muscle myoblasts HSkM were purchased from Gibco® (Thermo Fisher, A12555) and thawed in culture medium. HSkMs were then treated for 24 h with anti-IGF2R at 4 µg/ml. To induce myogenic differentiation in murine and human myoblasts, the culture medium was serum-depleted (final concentration, 2%), and the cells were harvested after 2, 4 and 6 days. ShCD20 (sc-29973-v) and scramble ShCTR (sc-108080) RNA particles were purchased from Santa Cruz Biotechnology. C2C12 cells were infected at a multiplicity of infection (MOI) of 10 in triplicate wells. Twenty-four hours post-transduction, fresh medium was added to each well, and the cells were maintained at 37°C in 5% $CO_2$. Infected cells were selected using puromycin (2 µg/ml) for 3 weeks, and silencing efficiency was evaluated by RT–PCR. The lentiviral vector used to induce CD20 over-expression was produced by GeneTarget Inc (USA). The target gene (mouse CD20/MS4a1 chr19:11,250,603–11,266,151) was obtained from a gene bank and subcloned into an expression lentivector using Eco cloning technology (a non-enzyme-based cloning technique). The target gene was expressed under the CMV early enhancer/chicken b-actin [CAG] promoter. The RFP-Puromycin reporter gene was expressed under a Rous sarcoma virus [Rsv] promoter. Thus, the target and reporter genes were independently expressed by two different promoters. Cells were infected ON at an MOI of 10. Infected cells were selected using puromycin linked to RFP. An empty vector was included as a control (see Appendix Fig S2).

### RT–PCR

To detect IGF2 mRNA, classical RT–PCR was carried out with 1 µg of cDNA as described in Benchaouir et al (2007) with primers listed in Appendix Table S1. After 36 cycles of amplification (94°C/2 min, 92°C/1 min, 66°C/2 min and 72°C/2 min), PCR products were analysed on 2% agarose gels.

### Sarcoplasmic reticulum isolation

The sarcoplasmic membrane was isolated in accordance with previously described methods (Parolini et al, 2009). Briefly, vastus medialis (VM) or tibialis anterior (TA) muscles were put in ice-cold homogenization buffer (250 mM sucrose and 5 mM HEPES pH 7 in 0.2% NaN₃) supplemented with complete protease inhibitor cocktail (Sigma-Aldrich) and homogenized with an electric homogenizer. The homogenates were then centrifuged at 5,500 g for 10 min at 4°C. The supernatants were harvested and centrifuged at 12,500 g for 18 min at 4°C. The pellets were discarded, and the supernatants were centrifuged again at 12,500 g for 18 min at 4°C. The supernatants were then transferred into ultracentrifuge tubes and centrifuged at 50,000 g for 1 h at 4°C. The pellets were resuspended in homogenization buffer supplemented with 600 mM KCl and complete protease inhibitor cocktail (Sigma-Aldrich) and incubated for 30 min on ice. The samples were then centrifuged at 15,000 g

for 10 min at 4°C. The resulting supernatants were centrifuged at 50,000 g for 1 h at 4°C. The final pellets were dissolved in homogenization buffer. The protein content was determined by standard BCA assays as in Parolini et al (2009).

### SERCA activity

Sarco/endoplasmic reticulum $Ca^{2+}$-ATPase activity was measured based on inorganic phosphate production using a commercially available kit (Nanjing Jiancheng Bioengineering Institute, China) according to the manufacturer's instructions. Endoplasmic reticulum (ER) fraction was obtained as described above, the protein concentration of the ER fraction was quantified, and the enzymatic activity of SERCA pumps was measured by an NADH-coupled assay (Warren et al, 1974; Strosova et al, 2011; Viskupicova et al, 2015). The SERCA-catalysed rate of ATP hydrolysis coupled to the oxidation of NADH was measured spectroscopically at a wavelength of 340 nm at PerkinElmer Wallac VICTOR2 Multi-label Counter 1420. Cell lysates were prepared in SERCA lysate buffer (250 mM sucrose, 5 mM HEPES pH 7, 0.2% NaN₃ supplemented with cOmplete Roche protease inhibitors); the lysate was used to measure SERCA activity. The SERCA preparation (final concentration 20 µg protein/well of 96 wells) was added to the assay mixture (100 mM KCl, 40 mM Hepes pH 7.2, 15 mM MgCl₂, 5.1 mM MgSO₄, 10 mM NaN₃, 5 mM phosphoenolpyruvate, 1 mM EGTA, 5 µM calcimycin, 0.15 mM NaDH, 1 mM CaCl₂, 20 U/ml pyruvate kinase, 20 U/ml lactate dehydrogenase). The reaction was started by addition of ATP (final concentration 5 mM). The reaction rate was determined spectroscopically by measuring the decrease in NADH absorbance at 340 nm, at 25°C for 20 min. Specific SERCA activity (IU/mg; i.e. µmol substrate/min/mg of protein) was calculated using the following equation:

$$\frac{IU}{mg} = \frac{\Delta A340 \; nm \times V}{6.22 \; m}$$

where $\Delta A340$ nm represent the variation of absorbance at 340 nm per min, V is the volume of the reaction mixture (mL), $6.22 \times 10^3$ L/mol/cm is the absorption coefficient for NADH, and m represents the total amount of protein in the reaction mixture (mg).

### Western blot (WB) analysis

To prepare total protein extracts, cells (C2C12, HSkM and 3T3), murine muscles (TA and VM muscles of C57BL and mdx mice) and biopsies of human biceps–brachialis form, healthy and DMD patients were homogenized with an electric homogenizer using RIPA buffer (0.1% SDS, 1% sodium deoxycholate, 1% Triton X-100, 50 mM Tris–HCl pH 8 and 150 mM NaCl) supplemented with complete protease inhibitor cocktail (Sigma-Aldrich) and PhosSTOP (Sigma-Aldrich) and then centrifuged at 18,900 g for 13 min. The supernatants were collected and stored at −20°C until analysed. For WB analyses of C2C12 cell lysates, cells were collected by trypsinization, washed twice with ice-cold PBS, lysed in NP-40 buffer (150 mM NaCl, 1% NP-40 and Tris–HCl 50 mM, pH 8.0) supplemented with complete protease inhibitor cocktail (Sigma-Aldrich) and PhosSTOP (Sigma-Aldrich) and then centrifuged at

18,900 $g$ for 13 min. Supernatants were collected and stored at −20°C until analysed. The total protein concentration was determined by standard BCA assays and quantified with a Glomax Microplate Luminometer (Promega, USA). The lysates were diluted in Laemmli sample buffer (4×) containing β-mercaptoethanol and boiled for 10 min. Total protein extracts and those obtained from isolated sarcolemma samples (see above) were separated on polyacrylamide gels and then transferred to supported nitrocellulose membranes (Bio-Rad Laboratories, USA). The filters were saturated in blocking solution (10 mM Tris pH 7.4, 154 mM NaCl, 1% BSA, 10% horse serum, 0.075% Tween-20). Primary antibodies were as follows: anti-CD20 (1:500; sc-7736 M-20, Santa Cruz), anti-Myogenin (1:250; cat. 556358, BD Biosciences, USA), anti-Myf5 (1:500; C-20 sc-302, Santa Cruz), anti-MyoD (1:1,000; 554130, BD Pharmingen), anti-NFAT (1:1,000; 4998, Cell Signalling, the Netherlands), anti-ORAI1 (1:1,000; SAB3500412, Sigma-Aldrich), anti-β-actin (1:1,000; A2066, Sigma-Aldrich), anti-IGF2R (1:500; sc-14413 K21, Santa Cruz), anti-GAPDH (1:1,000; V18 sc-47724, Santa Cruz), anti-CamKII (1:1,000; ab52476, Abcam), anti-pCAMKII Thr286 (1:1,000; MA1-047, Thermo Fisher, USA), anti-PKCα (1:1,000; 610107, BD Biosciences), anti-pPKCα Ser 657 (1:1,000; 06-822, Millipore, USA), anti-IGF1Rβ (1:200; sc-390130, Santa Cruz), anti-phospho-IGF1R (1:200; Tyr1161/Tyr1165/Tyr1166, ABE 332, Millipore), anti-Calcineurin A (1:500; ab3673, Abcam, United Kingdom), anti-pSer (1:500; 1C8 sc-81515, Santa Cruz), anti-pThr (1:500; H-2 sc-5267, Santa Cruz), anti-pIGF2R (1:500; ser2484, Thermo Fisher), anti-Gαi2 (1:400; sc-7276, Santa Cruz), anti-MyHC slow (1:500; M8421, Sigma-Aldrich), anti-SERCA1 (1:500; sc-515162, Santa Cruz), anti-CD146 (1:500; MEL-CAM C-20 sc-18492, Santa Cruz), anti-NG2 (1:500; sc-20162 H-300, Santa Cruz) and anti-vinculin (1:1,000; MA-11690, Sigma-Aldrich). The membranes were incubated with primary antibodies ON at 4°C, then followed by washing, detection with horseradish peroxidase (HRP)-conjugated secondary antibodies (DakoCytomation, USA) and developed by enhanced chemiluminescence (ECL) (Amersham Biosciences, USA). Prestained molecular weight markers (Bio-Rad Laboratories) were run on each gel. Bands were visualized using an Odyssey Infrared Imaging System (LI-COR Biosciences, USA). Densitometric analysis was performed using ImageJ software (http://rsbweb. nih.gov/ij/).

## Immunoprecipitation

Total protein extracts were prepared in 1% Nonidet P-40 detergent buffer (20 mM Tris pH 8, 137 mM NaCl, 2 mM EDTA, 10% glycerol and cOmplete® and PhosSTOP® cocktails, Roche). The antibodies used for immunoprecipitation of anti-CD20 (sc-7736 M-20, Santa Cruz), anti-ORAI-1 (ab59330, Abcam) and anti-IGF2R (sc-14413 K21, Santa Cruz) were cross-linked to 25 μl of Dynabeads Protein G (Thermo Fisher) in accordance with the manufacturer's instructions. Total protein lysates (1,000–1,500 μg) were incubated with the appropriate antibody–Dynabeads complexes and mixed ON at 4°C. The immune complexes were washed three times in 1 × PBS and the proteins were eluted boiling the complexes in 2 × Laemmli buffer (Tris 0.5 M pH 6.8; SDS 2%; glycerol 20%; β-mercaptoethanol; bromophenol blue) for 10 min. Next, the samples were centrifuged at 10,000 × $g$ for 5 min, and the resulting supernatants (IP samples) were collected.

## Immunofluorescence and immunohistochemistry analysis

CD20 was detected with anti-CD20 antibodies (M-20) (1:20) in accordance with Parolini et al (2012). Sarcomeric actin was detected using 647-phalloidin antibody (A22287 Alexa Fluor). IGF2 and IGF2R were detected using a rabbit and goat polyclonal antibody, respectively (1:100; sc-5622, H103 and sc-14413, K21, Santa Cruz). Nuclei were stained with 4′,6-diamidino-2-phenylindole (DAPI). Alexa Fluor 488- and 594-conjugated secondary antibodies (1:200, Abcam) were used. Images were captured using a Leica TCS SP2 confocal system (Leica, Germany). Muscles were removed, frozen in liquid nitrogen-cooled isopentane and then sectioned on a cryostat (LEICA CM 1850). Serial sections (8-μm-thick) were stained with haematoxylin and eosin and Azan-Mallory. Images were captured with a Leica DM6000B optical microscope. The TA and VM muscles were cut into longitudinal and transverse cryostat sections (10 μm) and characterized for their expression of endothelial and pericyte markers. The sections were fixed in 100% acetone or 80% ethanol and incubated for 2 h at room temperature (RT) with primary antibodies against CD31 (1:50; 550274 BD Biosciences), α-smooth muscle actin (1:400; A2547, Sigma-Aldrich), desmin (1:50; AB15200 Abcam), NG2 (1:100; AB5320 Chemicon), CD146 (1:50; Chemicon) and VE-cadherin (1:100; CD144 550548 eBioscience, USA). After the sections were washed with 1 × PBS, the slides were incubated with FITC- and Alexa Fluor 594-conjugated secondary antibodies (1:100; Thermo Fisher, USA) for 1 h at RT. After immunostaining was completed, the sections were counterstained with DAPI, coverslipped and examined by epifluorescence microscopy DMi8 (Leica). Fusion index was calculated as the ratio of the nuclei number of desmin-positive myotubes (≥ 2 nuclei) vs. the total number of nuclei analysed ($n$ > 300).

## Cytofluorimetric analysis and calcium imaging

Samples from three independent experiments were acquired with a FACS—Cytomics FC500 (Beckman Coulter, USA). For calcium imaging, C2C12 cells were incubated in DMEM containing 1.5 μM Fluo-4/AM supplemented with Pluronic-127 (Thermo Fisher) for 45 min at RT and then submitted to 45 min of de-esterification. Cells where then incubated with ERY $Ca^{2+}$ Ø for 15 min supplemented with 10 μM thapsigargin (Sigma-Aldrich), in 5 mM ERY $Ca^{2+}$ for 5 min and then analysed with the cytofluorimeter Cytomics FC500 (Beckman Coulter) or FACSCanto II (BD Biosciences) by cxp 2.1 software (BC) and FacsDiva software (BD Biosciences), respectively. In our experimental design, cells treated with 5 mM ionomycin for 5 min were included as a positive control. For calcium imaging, C2C12 cells were incubated in DMEM containing 1.5 μM Fluo-4/AM supplemented with Pluronic-127 (Thermo Fisher) for 45 min at RT, as previously described (Hamilton et al, 2008), followed by 45 min of de-esterification. Cells were then incubated with ERY $Ca^{2+}$ Ø (155 mM NaCl, 5 mM KCl, 1 mM $MgCl_2$, 10 mM HEPES) for 15 min, supplemented with 10 μM thapsigargin (Sigma-Aldrich) in 5 mM ERY $Ca^{2+}$ (155 mM NaCl, 5 mM KCl, 1 mM $MgCl_2$, 10 mM HEPES, 15 mM $CaCl_2$) for 5 min and, then, analysed with a cytofluorimeter Cytomics FC 500 (Beckman Coulter). In our experimental design, cells treated with 5 mM ionomycin for 5 min were included as a positive control. The same store depletion and staining protocol was followed for time-lapse analyses performed with a Nikon

BioStation IM (Nikon, Japan), where fluorescence images were collected every 4 min for 32 min. Thirty mM caffeine (Sigma-Aldrich) treatment was performed on standard depleted cells, and images were collected every 5 s for 4 min.

### Corrosion casting technique

Untreated and treated *mdx* mice (*n* = 10 per group) were sacrificed by a lethal dose of anaesthetic, and intraventricular infusion of 20 ml heparinized solution (19 ml of 0.9% NaCl solution and 1 ml of heparin) to prevent blood clotting and 10 ml of PBS (Phosphate Buffered Saline solution) to clear the remaining blood from the vascular bed and minimize the blood in the reservoirs was performed. Ten millilitres of an acrylic low-viscosity resin (MERCOX, SPI Supplies) was mixed with 0.2 ml of catalyser (benzoyl peroxide), diluted (20%) with methyl methacrylate (SPI Supplies) and then very slowly manually injected (1 ml/min) through a cannula inserted into the left heart ventricle, which was then closed with metallic staples to prevent any reflux of the resin. After the acrylic compound was allowed to partially polymerize for approximately 10 min, the animal was immersed in a warm water bath (60°C) for 1 h to complete the hardening process. Next, the VM or TA muscles were immersed in a 15% KOH solution to digest all the organic tissues around the vessels. This solution was changed every day for 1 week, during which the specimens were washed with distilled water supplemented with 5 ml of mycostatin to prevent fungal growth. The resulting casts were then dissected under a scanning electron microscopy (SEM) observations. They were dehydrated through a graded series of alcohol solutions, dried at critical point in an Emitech K850 CPD apparatus, mounted on aluminium stubs on adhesive film and coated with 10 nm of gold in an Emitech K250 Sputter Coater. Due to their size, some of the specimens needed to be fitted with metallic bridges to maintain good conduction throughout the stub. The specimens were then observed in a Philips XL-30 FEG SEM working at 10 kV. Quantitative evaluation of resin leakages from the vessels was performed at 250× in random fields. Data were then evaluated using common statistical analysis.

### Mechanics of isolated muscles

TA and VM muscle strips obtained from C57BL6 (*n* = 10) and either untreated (*n* = 10) or treated *mdx* (*n* = 10) mice were rapidly and carefully dissected, following the muscle fibre orientation, and placed in an organ bath filled with Krebs solution (120 mM NaCl, 2.4 mM KCl, 2.5 mM CaCl2, 1.2 mM MgSO$_4$, 5.6 mM glucose, 1.2 mM KH$_2$PO$_4$ and 24.8 mM NaHCO$_3$; pH 7.4). The bath was bubbled with 95% O$_2$ and 5% CO$_2$ at a constant temperature of 22°C and attached to a force transducer (Radnoti Organ Bath System, AD Instruments). Electrical pulses were delivered through platinum electrodes connected to a stimulator (Tumiati, Italy). Tetanic isometric contractions were evoked (110 Hz, 500 ms, supramaximal amplitude) at the length at which the maximal isometric force was observed (Lo), and the twitch time to peak and maximal tetanic force (Pt) values were measured. Muscle forces generated, including Pt and maximum tetanic force (Po), were normalized for the estimated physiological cross-sectional areas (CSA) of the muscle segment (CSA = muscle weight/1.056 × Lo; where 1.056 g/cm$^3$ represents the density of

muscle) and expressed in kilonewtons (kN)/m$^2$. All tests were performed in accordance with D'Antona *et al* (2007).

### Docking experiments

3D structure of IGF2R and epitope 3 of CD20 were retrieved from the Protein Data Bank (IGF2R-2v5p; epitopes 3 CD20-20sl and 3pp4). To investigate the molecular interactions of IGF2R, epitope 3 of CD20, chain A of 2v5p, chains P and Q of 2osl and chain P of 3pp4 were used in molecular docking studies. Docking simulations were performed using ClusPro 2.0 (Kozakov *et al*, 2017). Computational steps were performed as follows: rigid-body docking using the Fast Fourier Transform (FFT) correlation approach, root mean square deviation (RMSD) based on the clustering of structures to identify highly populated clusters that represented the most likely models of the complex, and refinement of selected structures using energy minimization. For each simulation, ClusPro returned clusters associated with low energy, and a representative sample was chosen by visual inspection and by selecting the most populated cluster that had the lowest energy.

### Assessment of plasma antibodies reactive against polyclonal goat anti-mouse IGF2R

Murine antibodies against polyclonal goat anti-mouse IGF2R titres were measured as previously described, with few modifications (Mazor *et al*, 2017). Cheek blood from mdx mice treated for 4 or 9 weeks with low- and high-dose anti-IGF2R antibody was collected into EDTA-treated tubes (Thermo Fisher Scientific). Samples were immediately centrifuged at 1,008 *g* for 5 min for collecting plasma and stored at −20°C. Plasma from not injected (n.i.) mdx mice was used as pre-dose control samples. Briefly, ELISA plates (Greiner Bio-One) were coated with 100 μl of 0.2 mg/ml goat anti-mouse IGF2R and stored overnight at 4°C. Before use, plates were rinsed extensively with 1× PBS with 0.05% Tween-20 (wash buffer) and then blocked with 5% BSA in 1× PBS with 0.05% Tween-20 (binding buffer) for 1 h at room temperature on a plate shaker. Serial dilutions of murine plasma samples were prepared in binding buffer starting with a 1:10 dilution. 100 μl of serially diluted samples was then added to the ELISA plates and incubated at room temperature on the shaker for 2 h. After incubation, the plates were washed three times with wash buffer and mouse antibodies against IGF2R were detected with anti-mouse IgG (H + L) HRP (Jackson Immuno-noResearch) (1:3,000 for 1 h at room temperature) and TMB ELISA substrate high sensitivity (Abcam). The reaction was stopped with H$_2$SO$_4$ stop solution, and optical density of the wells was read immediately at a wavelength of 450 nm with subtraction at 650 nm. Titres were calculated based on a four-parameter logistic curve-fit graph, and endpoint titre was calculated as the reciprocal of the highest dilution that gives signal above background absorbance. Results were expressed as fold changes over pre-dose values.

### Dynamic light scatter and nanotracking analysis evaluation of aggregates into antibody formulation

Dynamic light scatter (DLS) analysis was performed with Zetasizer Nano (Malvern), equipped with a standard laser (10 mW, 633 nm). This technique measures the diffusion of particles moving under

## Paper explained

### Problem

The progressive muscle weakness that affects Duchenne patients is determined by extremely low regenerative capacity, fibrosis, intracellular $Ca^{2+}$ de-regulation, inflammatory infiltrates and fibre necrosis. In the last years, different approaches have been investigated to treat DMD but, unfortunately, most therapeutic strategies have been palliative rather than curative. The influx of $Ca^{2+}$ into dystrophic muscle fibres is regulated by a plethora of mechanisms, such as membrane tears, stretch-activated channels and $Ca^{2+}$ leak channels. In addition, the function of SERCA, that is the main protein responsible for $Ca^{2+}$ reuptake into the sarcoplasmic reticulum, is compromised in murine model of DMD, the *mdx* mice. Members of the insulin-like growth factor (IGF) family are secreted during muscle repair, promoting muscle regeneration and hypertrophy. Since we found that IGF2R and the store-operated $Ca^{2+}$ channel CD20 share a common hydrophobic binding motif that reinforces their association, we wondered whether the modulation of IGF2R could have a therapeutic potential in DMD treatment.

### Results

We found that IGF2R expression was increased in muscle tissues of *mdx* mice, while the phosphorylation of IGF2R was significantly decreased. The targeting of IGF2R regulated the phosphorylation of CD20 and thereby induced SERCA activation in myoblasts. Interestingly, a delay in muscle differentiation was observed in CD20-silenced myoblasts, whereas the expression levels of early and late differentiation markers were increased after blockade of IGF2R. As these features were accompanied by the activation of the calmodulin/calcineurin/NFAT pathway, we proposed that an IGF post-translational modulatory mechanism regulates muscle differentiation. Following the blocking of IGF2R in *mdx* mice, we determined an increase in muscle regeneration and force, $Ca^{2+}$ reuptake and, more interestingly, an amelioration of blood vessels' architecture and maturation.

### impact

Our data strongly confirmed that IGF2R inhibition positively affected the regenerative potential of dystrophic muscle tissues. Overall, these evidences demonstrated that a biological therapy targeting IGF2R could lead to improvement of muscle regeneration and suppression of signalling cascades associated with pathological events in DMD muscles. The understanding of the contribution of aberrant IGF2R expression into the pathophysiology of DMD will pave the way to the development of novel therapeutic approaches.

Brownian motion and converts this to size and a size distribution using the Stokes–Einstein relationship. For measuring antibody formulation aggregates, 1 ml of 50 µg/ml polyclonal goat anti-mouse IGF2R diluted in 0.1-µm-filtered PBS1X was loaded into polystyrene cuvette. 0.1-µm-filtered 1 × PBS was used as control. Three measurements of each sample were performed, and the mean was obtained. NTA technique was performed with NanoSight NS300 (Malvern Instruments Limited, Worcestershire, UK). NTA measurement settings were set as follows: viscosity $0.91 \pm 0.03$ cP, temperature $23.75 \pm 0.5°C$, measurement time of 30 s for each of the five repeated records per sample, infusion flow speed 30 and camera level 14.

### Detection of anti-IGF2R antibody into injected muscles after treatment

In order to detect the presence of anti-IGF2R, we performed WB analysis of TAs and VMs of mdx mice injected for 4 and 9 weeks with low- and high-dose antibody. Muscles from n.i. mdx and C57Bl mice were used as controls. WBs analysis was performed as already described, with minimal modifications. Briefly, muscles were homogenized with an electric homogenizer in RIPA buffer supplemented with complete protease inhibitor cocktail and centrifuged to collect the supernatants. Protein extracts were separated on polyacrylamide gels and transferred to nitrocellulose membranes. After blocking, the membranes were incubated with HRP-conjugated rabbit anti-goat secondary antibodies (DakoCytomation, USA) and developed by ECL. Bands were visualized using Odyssey Infrared Imaging System.

### Statistical analysis

Values are expressed as the mean $\pm$ SD or SEM. In our case of 3 treatment groups (1: control; 2: low dose of antibody; and 3: high dose of antibody) with 10 animals per group, we used the Excel function = Rand() to generate a column of random numbers in column A and consequently in columns B and C. Blinding was maintained during and after the intervention, and the responsible of the experiments did not reveal the allocation sequence to the people that conducted the experiments until the data were analysed. Statistical comparisons among three or more groups were conducted using one- and two-way ANOVA followed by Bonferroni's multiple comparison test to determine significance (*$P < 0.05$, **$P < 0.01$, ***$P < 0.001$; ****$P < 0.0001$). Exact $P$-values can be found in Appendix Table S2. Otherwise, Student's $t$-test was applied.

**Expanded View** for this article is available online.

## Acknowledgements

This study was supported by Associazione Centro Dino Ferrari, an AFM Telethon grant (No. 21104), Ricerca Corrente 2019 (code: 230-01), Flagship "InterOmics" (cod. PB05), the MIUR PON CNRBiOMICS, AMANDA Project, CNR-Lombardy Agreement and Fondazione Regionale per la Ricerca Biomedica (LYRA_2015-0010-B92F16000670007) and by the Cariplo Foundation (2016-1006). This paper presents independent research funded by Roby and OPSIS Foundations. Prof. Torrente's laboratory has also received support from IRMI Project (Avviso n. 257/Ric. del 30 maggio 2012 – MIUR, project CTN01_00177_888744) and Ricerca Finalizzata 2016 (Research type: "Theory enhancing", project RF-2016-02362263).

## Author contributions

Conceptualization, YT; Methodology and Investigation, PB, SB, AF, DP, NT, MM, MB, SE, PD, FB, CR, CS, SS, CV, GDA and ML; Data analysis and discussion, YT, EN, PM and LM; writing original draft, review and editing, EN, PD, PB, AF and CV with inputs from all authors; and final manuscript main funding and coordination, YT.

## Conflict of interest

The authors declare that they have no conflict of interest.

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
