## [Review Process File · EMBO Molecular Medicine]

Blockade of IGF2R improves muscle regeneration and ameliorates Duchenne muscular dystrophy

Pamela Bella, Andrea Farini, Stefania Banfi, Daniele Parolini, Noemi Tonna, Mirella Meregalli, Marzia Belicchi, Silvia Erratico, Pasqualina D'Ursi, Fabio Bianco, Mariella Legato, Chiara Ruocco, Clementina Sitzia, Simone Sangiorgi, Chiara Villa, Giuseppe D'antona, Luciano Milanesi, Enzo Nisoli, PierLuigi Mauri, Yvan Torrente

Review timeline:

Submission date:	18 June 2019
Editorial Decision:	17 July 2019
Revision received:	12 September 2019
Editorial Decision:	2 October 2019
Revision received:	29 October 2019
Accepted:	30 October 2019

Editor: Céline Carret

Transaction Report:

1st Editorial Decision

17 July 2019

Thank you for the submission of your manuscript to EMBO Molecular Medicine. We have now heard back from the two referees whom we asked to evaluate your manuscript. We initially had a 3rd referee lined up, but this referee dropped out unfortunately, so we prefer to make a decision now and not involved yet another person that would considerably delay publication.

You will see that both reports are encouraging but highlight issues with the study that must be addressed in a major revision of this work. Ex vivo and in vivo additional experiments have to be performed to increase significance. Proper controls have to be provided. Details / explanations / clarifications are also needed throughout the article.

We would therefore welcome the submission of a revised version within three months for further consideration and would like to encourage you to address all the criticisms raised as suggested to improve conclusiveness and clarity. Please note that EMBO Molecular Medicine strongly supports a single round of revision and that, as acceptance or rejection of the manuscript will depend on another round of review, your responses should be as complete as possible.

I look forward to receiving your revised manuscript.

***** Reviewer's comments *****

Referee #1 (Comments on Novelty/Model System for Author):

Technical quality: Small sample size (n=3 mice) in some studies is inadequate. Using beta-actin which varies in muscle disease to normalise for loading is a concern.

Novelty: The study is of medium novelty based on previous studies

Medical Impact: Therapy could be developed using this study but appropriate antibodies would need to be developed so impact would be in the future.

Model System: The mdx mouse model is adequate. C2C12 cells are okay but human myoblasts should be included to confirm the mouse myoblast studies.

Referee #1 (Remarks for Author):

This manuscript investigates the role insulin growth factor receptor 2 (IGF2R) plays in muscle regeneration and muscular dystrophy. The manuscript describes IGF2R is increased in muscle from DMD patients and the mdx mouse model. Blockade of this receptor using a neutralizing antibody induced CD20 phosphorylation, activation of SERCA in myogenic cells and enhanced muscle regeneration, muscle force recovery and muscle vasculature.

There are several concerns that need to be addressed:

Major:

1. There is no statement on the use of human subjects in the study in the methods section. Please indicate if these tissues were obtained using informed consent or was off the shelf tissues and therefore exempt.
2. How many patients were used in the study? Is the tissue from similar aged DMD and control patients?
3. There is no information on when treatment was started using mdx mice in the study. Mice were treated from 4-9 weeks starting at what age?
4. Please provide information on the anti-IGF2R antibody used from R&D Systems. Was it an anti-human or mouse monoclonal or polyclonal and please provide catalogue information?
5. Fig 2G: the anti-IGF2R fusion index is increased compared to untreated. Some comment on the mechanism by which this improved fusion would be relevant
6. How were the low and high antibody concentrations used to treat mice determined? What is the pharmacokinetics of the antibody in serum and muscle?
7. Fig 3E: Wild-type mouse muscle served as controls for DMD patient samples. Due to species differences, this is not an appropriate control for the human studies. Studies using human DMD muscle should be compared to unaffected patients muscles.
8. Fig 4C: The high anti-IGF2R at 9 weeks the western blot lacks a MHC band, yet the graph shows high levels of MHC. A rationale for this apparent inconsistency needs to be provided.
9. Fig 4E: Are the ATP hydrolysis data statistically different? In vivo or ex vivo muscle force measurements should be performed to confirm and quantify improve muscle strength.
10. Fig 6: only an n=3 mice were used. This small sample size could result in statistical errors and sample size should be increased.
11. Fig 7A: Muscle from 4 and 9 week old wild-type mice should be included to compare vascular beds with untreated and anti-IGF2R treated animals.
12. Throughout the study beta-actin is used as a loading control. It is well known that dystrophic muscle disease affects changes in actin levels which will also vary. Using the intensity of all bands (using Ponceau S staining or other total protein staining methods) is now considered a more accurate method of normalising for band intensity.
13. A model of DMD muscle disease with and without IGF2R treatment and regulation of signaling pathways identified in the study would benefit the manuscript
14. Scale bars should be included in all images. Fig 3A & B; Fig 7A & B lack scale bars

Minor

1. Remove the duplicated "in" within the abstract
2. In some places, the manuscript is rather dense and difficult to follow. Please try and simplify the manuscript in these places to make it easier for the reader to follow e.g. splitting Figure 1 into 2 separate Figures to make it easier for the reader.

Referee #2 (Remarks for Author):

Bella et al. explored the role of IGF2R/CD20 binding in C2C12 myoblast differentiation; the inhibition of IGF2R in mdx mice shows improved muscle regeneration and muscle tetanic force via SERCA activation and Ca²⁺ re-uptake and ameliorated vascular network. This is an interesting piece of work and the finding could lead to the development of a novel treatment for DMD.

- Graphs and writing style need to be edited.

Minor comments:

Page 3, Results, line 3 "IGF1 and IGF1R expression were not increased in IGF1-treated C2C12 myoblasts" These results must be included in Fig 1A as part of the same assay.

Fig 1B Control data and labels are missing.

Fig 1C Re-adjust the last two labels.

Fig 1 legend Concentrations should be in the methods and not in the legend.

Fig 1E The anti-CD20 WB looks blurry. Quantification cannot be conclusive and the WB assay has to be repeated.

Fig 2 legend The IGF2R domain 11 AB, CD is orange/red and not yellow.

Fig 2E Missing labels.

Fig 2G/H G is part of F. H has to be moved next to the WB. There are some concerns regarding the MyHC expression in the last WB sample; it peaks at day 4 but not at day 6 in contrast with the image showing large myotubes (also the b-actin control seems lower).

Fig EV2C The gating strategy from plot 2 to 3/4 is not clear. Double positive 91.2% or 32%?

Fig EV3J The WB has to be repeated.

Fig EV3 H to L content should be in Fig EV4 and the EV4 content in Fig EV3.

Fig 3 Where is the laminin staining (green) reported in the legend? Is this needed?

Fig 3B The IGF2R staining seems non-specific in the DMD2 patient.

Fig 3D Missing in the figure the label D.

Fig 3F Control must be included.

Page 7 How the IGF2R antibody dosage has been selected? How many injections?

Fig 6A should have only A and B. Quantifications refer to the WBs.

Page 8 Fig 6B should be part of Fig 6A, the current Fig 6C/D should be Fig 6B.

Fig 6A What happens to the CD20 expression (low dosage, 9weeks) in VM?

Fig 7A The WT control sample is missing. TAs or VMs in A?

Fig 7D should be C.

Fig 7C Comparison with controls is missing.

Major:

Fig EV3 How does the author explain the discrepancy between untreated and anti-IGF2R samples in F and G?

Page 6 "We found that the level of CD20 phosphorylation was higher in mdx muscle and that this change was related to an alteration in IGF homeostasis" Where are the data supporting this statement?

Fig 4D TAs: the tetanic force is low in the control mice compared to the anti-IGF2R treated (4weeks), at 9weeks there are opposite results and no force improvements. How can this inconsistency be explained? VMs: tetanic force variability at 4 weeks and improvements at 9 weeks. How can the variation between muscles be explained?

Fig 4E TAs: the ATP hydrolysis (4weeks) shows no correlation with D (4weeks). VMs: the low dose is missing.

Fig 6C/D The disparity observed between muscles, between control and experimental samples could mislead the data interpretation and conclusions.

1st Revision - authors' response

12 September 2019

Referee #1 (Remarks for Author):

This manuscript investigates the role insulin growth factor receptor 2 (IGFR2) plays in muscle regeneration and muscular dystrophy. The manuscript describes IGFR2 is increased in muscle from DMD patients and the mdx mouse model. Blockade of this receptor using a neutralizing antibody induced CD20 phosphorylation, activation of SERCA in myogenic cells and enhanced

muscle regeneration, muscle force recovery and muscle vasculature.

There are several concerns that need to be addressed:

Major:

1. There is no statement on the use of human subjects in the study in the methods section. Please indicate if these tissues were obtained using informed consent or was off the shelf tissues and therefore exempt.

The use of animals in this study was authorized by the National Ministry of Health (protocol number 10/13-2014/2015). Muscle biopsies were obtained from the Telethon biobank (GTB12001) and used for research procedures that were approved by the ethics committee of the University of Milan (CR937-G). This is now specified in the M&M section.

How many patients were used in the study? Is the tissue from similar aged DMD and control patients?

Two DMD patients of 10 and 11-years old and two healthy donors of 19 and 22 years-old have been selected for this study. This is now better explained in the M&M section.

3. There is no information on when treatment was started using mdx mice in the study. Mice were treated from 4-9 weeks starting at what age?

All the animals have been treated at 3 months old as mentioned in Results section and now included in the M&M section.

4. Please provide information on the anti-IGF2R antibody used from R&D Systems. Was it an anti- human or mouse monoclonal or polyclonal and please provide catalogue information?

We used the Polyclonal Goat IgG anti IGF2R (AF2447, R&D Systems). This is now better specified in the M&M section.

5. Fig 2G: the anti-IGF2R fusion index is increased compared to untreated. Some comment on the mechanism by which this improved fusion would be relevant

The fusion index was calculated as the ratio of the nuclei number in myocytes with two or more nuclei versus the total number of nuclei. We have attempted to evaluate the fusion index to evaluate the differentiation of myoblasts in mature myotubes. We now provide a model to resume the underlying mechanisms of the increased differentiation of anti-IGF2R treated myoblasts and included this model in a Synopsis. In summary, during exercise, Ca^{2+} is cycled between the cytosol and the sarcoplasmic reticulum (SR) through a system by which the Ca^{2+} pool in the SR is restored by uptake of extracellular Ca^{2+} via a mechanism called store-operated Ca^{2+} entry (SOCE). Muscular dystrophies (MD) are often associated with Ca^{2+} dyshomeostasis. Treatment with anti-IGF2R activate IGF2R- $\text{G}\alpha\text{i}2$ interactions and CD20 phosphorylation promoting the entrance of Ca^{2+} ions in the sarcoplasm. Blockade of IGF2R facilitates IGF2-IGF1R interactions with consequent IGF1R phosphorylation which recruits the PI3-K/Akt/mTOR signaling axis to regulate the expression of skeletal muscle-specific genes that are associated with myogenic differentiation. PI3-K regulates the CD20 phosphorylation. Increasing levels of Ca^{2+} ions in the sarcoplasm bind to and activate calmodulin (CaM) which regulates activation of calcineurin and calmodulin kinase II (CaMKII). Calcineurin dephosphorylates nuclear factor of activated T-cells (NFAT) hereby regulating its nuclear localization and facilitate the increased expression of genes involved in muscle regeneration and capillary remodeling. CaMKII activate the reuptake of Ca^{2+} ions in the SR regulating the ATP-dependent sarcoplasmic/endoplasmic reticulum calcium ATPase pump (SERCA1). Intracellular Ca^{2+} ions critically regulates contraction and force production of muscle fibers. Free Ca^{2+} ions also directly stimulate or inhibit Ca^{2+} release via RyR1 in dependency of their luminal and sarcoplasmic Ca^{2+} concentration. Declined Ca^{2+} ion concentrations in the SR activate interaction of STIM1 with Orai1 leading to trans-sarcolemmal Ca^{2+} influx. Blockade of IGF2R reestablished the correct oscillating pattern of Ca^{2+} ion levels in the microenvironment of myofibrils and protects from MD acting on different mechanisms of dystrophic muscles.

6. How were the low and high antibody concentrations used to treat mice determined? What is the pharmacokinetics of the antibody in serum and muscle?

The function of the cation independent mannose 6-phosphate/insulin-like growth factor 2 receptor (IGF2R) is in the transportation and regulation of the extra-cellular bioavailability of Insulin-like growth factor 2 (IGF2) and mannose 6-phosphate modified proteins¹. The homology between

human and mouse IGF2R is 81% and its domain 11 is highly conserved across all vertebrate species. To determine the range of dosages of anti-IGF2R, we treated animals daily intravenously. The effects of the IGF2R blockade were assayed considering: 1) plasma levels of IGF2 in wt (1ng/ml) and mdx (0.5ng/ml) and circulating IGF2R in wt (0.16ng/ml) and mdx (20ng/ml)(manuscript in preparation) and 2) that a previous in vitro competition assay showed that anti-IGF2R (R&D Systems) at the concentration of 20 µg/mL blocks >90% of the binding between Recombinant Human IGF-II/IGF2 (50 ng/mL) and the Recombinant Human IGF-II R/IGF2R (2 µg/mL) (IGF2/IGF2R 1:40) (see applications and general protocols of the R&D Systems web site for the Recombinant Human IGF-II R/IGF2R, Catalog # 2447-GR). Based on this premises, we set a range of antibody concentration ranging from 10 to 100 molar excess with respect to the plasma levels of IGF2. On average, mice have around 58.5 ml of blood per kg of bodyweight. We tested mouse weighing 20 g having a total blood volume (TBV) of approximately 58.5 ml/kg x 0.020 kg = 1.17 ml. Thus we selected a range of IGF2R blockade dosages between 10 and 100mg/ per mouse of 20gr weight. IGF2R Blockade doses were estimated to achieve at least 100-fold molar excess over IGF2R assuming that concentrations of the latter would reach peak values of 10 ng/ml of plasma, that is, values comparable to those detected in the mdx. A 100-fold molar excess of anti IGF2R over IGF2 specifically produces more than 95% inhibition of the binding of IGF2 in muscle culture assays. We tested the pharmacokinetics of the anti-IGF2R delivered intravenously and found that the decay of the anti-IGF2R in serum of treated animals is around 18 hours suggesting daily administration as optimal treatment. All these data support a different set of experimental evidences contained in a separate manuscript in preparation evaluating the agonist or antagonist effects of IGF2R binding peptides in mdx mice.

7. Fig 3E: Wild-type mouse muscle served as controls for DMD patient samples. Due to species differences, this is not an appropriate control for the human studies. Studies using human DMD muscle should be compared to unaffected patients' muscles.

We appreciate this comment and included healthy human muscles in these experiments. The new Figure 3 shown healthy human muscle as control for DMD and C57 wild type murine muscle as control for the mdx.

8. Fig 4C: The high anti-IGF2R at 9 weeks the western blot lacks a MHC band, yet the graph shows high levels of MHC. A rationale for this apparent inconsistency needs to be provided.

In agreement with this referee we included new representative WB of MHC and relative housekeeping in the revised Figure 4C.

9. Fig 4E: Are the ATP hydrolysis data statistically different? In vivo or ex vivo muscle force measurements should be performed to confirm and quantify improve muscle strength.

Statistical analysis of ATP hydrolysis have been previously reported in the results section as “ATP hydrolysis was significantly increased in the TA ($p < 0.0001$ for the first 3 minutes and $p < 0.01$ after 4 and 5 minutes for low dosages after 4 weeks; $p < 0.05$ for the first minute for high dosages after 9 weeks) and VM ($p < 0.0001$ for the first 4 minutes, $p < 0.001$ after 5 minutes, $p < 0.01$ after 6 minutes and $p < 0.05$ after 7 minutes for high dosages after 4 weeks, two-way ANOVA with Bonferroni correction) muscles of 4 and 9 weeks low dose anti-IGF2R-treated mice than in untreated mdx mice (Fig 4E).” In the previous version of the manuscript we included the muscle force measurements (Fig 4D) expressed as ex vivo tetanic muscle force. However, we understand that it was not clear thus we improved the presentation of the data in the revised Results section.

10. Fig 6: only an n=3 mice were used. This small sample size could result in statistical errors and sample size should be increased.

In agreement with this referee we performed more experiments to increase the sample size to n=10 mice per groups. We thank this referee for this comment because we improved our statistical analysis reducing the SD between specimens.

11. Fig 7A: Muscle from 4 and 9 week old wild-type mice should be included to compare vascular beds with untreated and anti-IGF2R treated animals.

We thank this reviewer for this comment. We included vascular analysis of three months old C57Bl wild type mice and provided new images in Figure 7.

12. Throughout the study beta-actin is used as a loading controls. It is well known that dystrophic muscle disease affects changes in actin levels which will also vary. Using the intensity of all bands

(using Ponceau S staining or other total protein staining methods) is now considered a more accurate method of normalising for band intensity.

The alpha-actin could be involved in muscle disease since is one of the players of the muscle contraction and cytoskeleton structure. Beta and gamma forms are the only two non-muscle members. They play important roles in basic cellular processes, and are expressed in all cell types. Beta-actin is the most commonly used loading control for western blot and several authors previously reported this housekeeping in DMD animal models²⁻⁵ and DMD patients⁶⁻⁸. For these reasons we selected the beta-actin as loading control. Although we agree that Ponceau could be representative of total proteins (and now considered more accurate), unfortunately we don't have the chance to retrieve the original Ponceau of all the gels or to repeat all the WB experiments. Finally, we never observed beta-actin modifications from our wb gels performed in triplicate from mdx mice tested in this paper and from other experiments performed in our lab.

13. A model of DMD muscle disease with and without IGF2R treatment and regulation of signaling pathways identified in the study would benefit the manuscript

As suggested we included in the new version of the manuscript a Synopsis with the model of the pathway related to the anti-IGF2R treatment.

14. Scale bars should be included in all images. Fig 3A & B; Fig 7A & B lack scale bars

Scale bars have been included in all the figures.

Minor

1. Remove the duplicated "in" within the abstract

The duplicate "in" has been removed from the abstract.

2. In some places, the manuscript is rather dense and difficult to follow. Please try and simplify the manuscript in these places to make it easier for the reader to follow e.g. splitting Figure 1 into 2 separate Figures to make it easier for the reader.

As suggested we revised our manuscript separating the results sections in paragraphs and re-editing all the figures.

Referee #2 (Remarks for Author):

Bella et al. explored the role of IGF2R/CD20 binding in C2C12 myoblast differentiation; the inhibition of IGF2R in mdx mice shows improved muscle regeneration and muscle tetanic force via SERCA activation and Ca²⁺ re-uptake and ameliorated vascular network. This is an interesting piece of work and the finding could lead to the development of a novel treatment for DMD.

- Graphs and writing style need to be edited.

All graphs and writing style have been edited as suggested.

Minor comments:

Page 3, Results, line 3 "IGF1 and IGF1R expression were not increased in IGF1-treated C2C12 myoblasts" These results must be included in Fig 1A as part of the same assay.

In agreement with this referee we included the RT-PCR analysis for IGF1 and IGF1R-b expression of untreated and IGF1-treated C2C12 myoblasts (new Figure 1A). These results are now described in the text.

Fig 1B Control data and labels are missing.

The IF staining for IGF2 expression in untreated C2C12 myoblasts as well as labels have been now included in the new Figure 1B.

Fig 1C Re-adjust the last two labels.

Labels of Figure 1C have been corrected.

Fig 1 legend Concentrations should be in the methods and not in the legend.

We deleted from the legend of Figure 1 the redundant concentrations and maintained only those needed for the understanding.

Fig 1E The anti-CD20 WB looks blurry. Quantification cannot be conclusive and the WB assay has to be repeated.

To sustain our quantifications we included in the new Figure 1E the representative images of WB for anti-CD20 and anti pThr/pSer.

Fig 2 legend The IGF2R domain 11 AB, CD is orange/red and not yellow.

Yellow correspond to the IGF2. This is now better indicated in the legend of Figure 2.

Fig 2E Missing labels.

Labels are now inserted in the new Figure 2.

Fig 2G/H G is part of F. H has to be moved next to the WB. There are some concerns regarding the MyHC expression in the last WB sample; it peaks at day 4 but not at day 6 in contrast with the image showing large myotubes (also the b-actin control seems lower).

Figure 2 was modified as suggested. The curves of WB for MyHC expression showed an anticipation of the myogenic differentiation reason why the amount of MyHC protein is increased at day 4 and decreased at day 6. The amount of MyHC protein depends on the phenotype of the myotubes. To verify these hypotheses, we monitored other parameters of differentiation progression: the length and thickness of myotubes and the number of nuclei per fiber. Moreover, IGF2R blockade-induced myotubes were found to be longer and to have a low number of nuclei per fiber compared to untreated myotubes which appeared larger with a high number of nuclei per fiber (Fig 2F), indicating that muscle differentiation of myoblasts exposed to IGF2R blockade proceeded for those cells that started prematurely to fuse. We included these observations in the Results section and commented as “We reasoned that this could depend either on a premature differentiation of anti-IGF2R treated C2C12 myoblasts or on the presence of a mixed population of proliferating and differentiating cells”.

Fig EV3C The gating strategy from plot 2 to 3/4 is not clear. Double positive 91.2% or 32%?

In order to clarify this point we modified Fig EV3C deleting plot 3 and 4 and indicating in the text the Fluo4/CD20 double positive (32%) and the Fluo4-/CD20+ cells (57.4%).

Fig EV3J The WB has to be repeated.

The WB was repeated as suggested and inserted in the new version of EV4C.

Fig EV3 H to L content should be in Fig EV4 and the EV4 content in Fig EV3.

As suggested we modified the EV3 and EV4 figures.

Fig 3 Where is the laminin staining (green) reported in the legend? Is this needed?

Laminin was a mistake now corrected in the legend of Figure 3.

Fig 3B The IGF2R staining seems non-specific in the DMD2 patient.

We now included a new image for DMD 2 in the revised Figure 3B

Fig 3D Missing in the figure the label D.

Label D is now provided in the Figure 3.

Fig 3F Control must be included.

Human healthy controls have been included in Figure 3 F.

Page 7 How the IGF2R antibody dosage has been selected? How many injections?

The function of the cation independent mannose 6-phosphate/insulin-like growth factor 2 receptor (IGF2R) is in the transportation and regulation of the extra-cellular bioavailability of Insulin-like growth factor 2 (IGF2) and mannose 6-phosphate modified proteins¹. The homology between human and mouse IGF2R is 81% and its domain 11 is highly conserved across all vertebrate species. To determine the range of dosages of anti-IGF2R, we treated animals daily intravenously. The effects of the IGF2R blockade were assayed considering: 1) plasma levels of IGF2 in wt (1ng/ml) and mdx (0.5ng/ml) and circulating IGF2R in wt (0.16ng/ml) and mdx (20ng/ml)(manuscript in preparation) and 2) that a previous in vitro competition assay showed that anti-IGF2R (R&D Systems) at the concentration of 20 µg/mL blocks >90% of the binding between Recombinant

Human IGF-II/IGF2 (50 ng/mL) and the Recombinant Human IGF-II R/IGF2R (2 µg/mL) (IGF2/IGF2R 1:40) (see applications and general protocols of the R&D Systems web site for the Recombinant Human IGF-II R/IGF2R, Catalog # 2447-GR). Based on this premises, we set a range of antibody concentration ranging from 10 to 100 molar excess with respect to the plasma levels of IGF2. On average, mice have around 58.5 ml of blood per kg of bodyweight. We tested mouse weighing 20 g having a total blood volume (TBV) of approximately 58.5 ml/kg x 0.020 kg = 1.17 ml. Thus we selected a range of IGF2R blockade dosages between 10 and 100mg/ per mouse of 20gr weight. IGF2R Blockade doses were estimated to achieve at least 100-fold molar excess over IGF2R assuming that concentrations of the latter would reach peak values of 10 ng/ml of plasma, that is, values comparable to those detected in the mdx. A 100-fold molar excess of anti IGF2R over IGF2 specifically produces more than 95% inhibition of the binding of IGF2 in muscle culture assays. We tested the pharmacokinetics of the anti-IGF2R delivered intravenously and found that the decay of the anti-IGF2R in serum of treated animals is around 18 hours suggesting daily administration as optimal treatment. All these data support a different set of experimental evidences contained in a separate manuscript in preparation evaluating the agonist or antagonist effects of IGF2R binding peptides in mdx mice.

Fig 6A should have only A and B. Quantifications refer to the WBs. Page 8 Fig 6B should be part of Fig 6A, the current Fig 6C/D should be Fig 6B.

Figure 6 was modified as suggested and relative results corrected in the text.

Fig 6A What happens to the CD20 expression (low dosage, 9weeks) in VM?

We performed more WB analysis to be consistent with our data and increase statistical differences. All WB images of the previous Figure 6 have been replaced with the new ones and statistical analysis performed using higher number of animals (n=10 per group).

Fig 7A The WT control sample is missing. TAs or VMs in A?

As suggested we included the WT control image in A. In previous legend we already specified the VM muscle origin of images.

Fig 7D should be C.

Figure 7 has been completely reformatted.

Fig 7C Comparison with controls is missing.

As suggested we included in the new figure 7 a representative image of the control staining of untreated mdx.

Major:

Fig EV3 How does the author explain the discrepancy between untreated and anti-IGF2R samples in F and G?

The effective phase of SOCE was impaired in CD20-silenced C2C12 myoblasts treated with anti-IGF2R antibodies, suggesting that CD20 is activated by anti-IGF2R during the store depletion (SD)(Fig EV3F). The caffeine-induced intracellular Ca^{2+} peak was almost completely abolished in untreated and CD20-silenced cells, demonstrating that SR stores are effectively emptied (Fig EV3G). In contrast, a Ca^{2+} peak was clearly observed in cells exposed to anti-IGF2R antibodies indicating that SD efficiency was reduced by blockade of IGF2R via the activation of SERCA1.

Page 6 "We found that the level of CD20 phosphorylation was higher in mdx muscle and that this change was related to an alteration in IGF homeostasis" Where are the data supporting this statement?

We corrected this sentence as "We found that the level of CD20 phosphorylation was higher in mdx muscle and that this change was related to an alteration in IGF2R expression".

Fig 4D TAs: the tetanic force is low in the control mice compared to the anti-IGF2R treated (4weeks), at 9weeks there are opposite results and no force improvements. How can this inconsistency be explained? VMs: tetanic force variability at 4 weeks and improvements at 9 weeks. How can the variation between muscles be explained?

The tetanic force is expressed as Po/CSA (kN/m²) which mean that the force is normalized to muscle fiber cross sectional area. The treated mdx have an increased muscle regeneration with increased number of fibers with reduced CSA (see Figure 4A) and similar force to wild type mice

leading to 150% improvement of treated mdx compared to wild type animals. This point is now commented in the results sections as “To confirm these findings, we verified that muscle function was recovered in the IGF2R-treated mdx mice and evaluated the maximum tetanic force (Po) normalized for the cross-sectional areas (CSA). We found that specific force, was ameliorated in the anti-IGF2R-treated mdx mice ($p < 0.05$, $p < 0.001$ two-way ANOVA with Bonferroni correction) (Fig 4D). Interestingly, the specific force generated by TA muscles of mice treated with both low and high doses of anti-IGF2R and VM muscles of mice treated with low dose of anti-IGF2R for 4 weeks, was higher than TA and VM muscles of control C57Bl6/J (Fig 4D). The increased specific force and the reduction of CSA of these anti-IGF2R cohorts indicate similar values of Po between anti-IGF2R treated mdx and C57Bl6J control mice”. Moreover, differences in muscle vasculature may influence muscle strength variability between anti-IGF2R treated muscles (see Figure 7).

Fig 4E TAs: the ATP hydrolysis (4weeks) shows no correlation with D (4weeks). VMs: the low dose is missing.

We thank this referee for this comment and inserted the missing low dose values in the new Figure 4E.

Fig 6C/D The disparity observed between muscles, between control and experimental samples could mislead the data interpretation and conclusions.

In agreement with this referee we have completely redone the figure 6 reporting new WB experiments and statistical analysis.

Cited References

- 1 Ghosh, P., Dahms, N. M. & Kornfeld, S. Mannose 6-phosphate receptors: new twists in the tale. *Nature reviews. Molecular cell biology* **4**, 202-212, doi:10.1038/nrm1050 (2003).
- 2 Amirouche, A. *et al.* Converging pathways involving microRNA-206 and the RNA-binding protein KSRP control post-transcriptionally utrophin A expression in skeletal muscle. *Nucleic acids research* **42**, 3982-3997, doi:10.1093/nar/gkt1350 (2014).
- 3 Kornegay, J. N. *et al.* Dystrophin-deficient dogs with reduced myostatin have unequal muscle growth and greater joint contractures. *Skeletal muscle* **6**, 14, doi:10.1186/s13395-016-0085-7 (2016).
- 4 Peladeau, C., Adam, N. J. & Jasmin, B. J. Celecoxib treatment improves muscle function in mdx mice and increases utrophin A expression. *FASEB journal : official publication of the Federation of American Societies for Experimental Biology* **32**, 5090-5103, doi:10.1096/fj.201800081R (2018).
- 5 Zhao, W., Wang, X., Sun, K. H. & Zhou, L. alpha-smooth muscle actin is not a marker of fibrogenic cell activity in skeletal muscle fibrosis. *PLoS One* **13**, e0191031, doi:10.1371/journal.pone.0191031 (2018).
- 6 Kameyama, T. *et al.* Efficacy of Prednisolone in Generated Myotubes Derived From Fibroblasts of Duchenne Muscular Dystrophy Patients. *Frontiers in pharmacology* **9**, 1402, doi:10.3389/fphar.2018.01402 (2018).
- 7 Korner, H. *et al.* Digital karyotyping reveals frequent inactivation of the dystrophin/DMD gene in malignant melanoma. *Cell cycle* **6**, 189-198, doi:10.4161/cc.6.2.3733 (2007).
- 8 Vianello, S. *et al.* SPP1 genotype and glucocorticoid treatment modify osteopontin expression in Duchenne muscular dystrophy cells. *Human molecular genetics* **26**, 3342-3351, doi:10.1093/hmg/ddx218 (2017).

2nd Editorial Decision

2 October 2019

Thank you for the submission of your revised manuscript to EMBO Molecular Medicine. We have now received the enclosed reports from the referees that were asked to re-assess it. As you will see the reviewers are now globally supportive. However, before we move forward we would like to encourage you to address the last set of comments from referee #1.

You will see that while referee 1 initially asked to confirm the main results using human myoblast cells, this was not done and this referee asks again for it. In addition, this referee has questions that need answering and the answers should be reflected in the final article.

***** Reviewer's comments *****

Referee #1 (Comments on Novelty/Model System for Author):

Technical Quality: Has been improved by increasing animal numbers

Novelty: The study is of medium novelty based on previous studies

Medical Impact: Therapy could be developed using this study but appropriate antibodies would need to be developed so impact would be in the future.

Model System: The mdx mouse model is adequate. C2C12 cells are okay but human myoblasts should be included to confirm the mouse myoblast studies.

Referee #1 (Remarks for Author):

This revised manuscript investigates the role insulin growth factor receptor 2 (IGFR2) plays in muscle regeneration and muscular dystrophy.

The authors have addressed many of my concerns and greatly, however there are a few questions that need further clarification:

1. Did the authors observe an immune response to the goat polyclonal anti-IGF2R antibody in the mouse?
2. Could a mouse immune response to the goat polyclonal anti-IGF2R antibody complicate data interpretation?
3. How much of the anti-IGF2R antibody is present in the muscle of mdx mice after 9 weeks of treatment?
4. Is the treatment antibody at steady state in the muscle or do levels fluctuate with treatment?
5. What is the proposed mechanism of IGF2R blockade using this antibody e.g. blocking the ligand-receptor interaction or receptor internalization and loss or some other mechanism(s)?

Referee #2 (Remarks for Author):

All suggested revision points have been properly and satisfactorily addressed which make the manuscript ready for publication.

2nd Revision - authors' response

29 October 2019

Point by point response to Referee #1

Technical Quality: Has been improved by increasing animal numbers

Novelty: The study is of medium novelty based on previous studies

Medical Impact: Therapy could be developed using this study but appropriate antibodies would need to be developed so impact would be in the future.

Model System: The mdx mouse model is adequate.

C2C12 cells are okay but human myoblasts should be included to confirm the mouse myoblast studies.

According to referee's request, we now added human myoblast results in Fig 2 and in the main text. Human skeletal myoblasts were treated with anti-IGF2R and tested for their myogenic capacity. The immunofluorescence, fusion index and WB analysis of early and late myogenic differentiation markers are now reported in the text and in Figure 2.

This revised manuscript investigates the role insulin growth factor receptor 2 (IGFR2) plays in muscle regeneration and muscular dystrophy. The authors have addressed many of my concerns and greatly, however there are a few questions that need further clarification:

- 1. Did the authors observe an immune response to the goat polyclonal anti-IGF2R antibody in the mouse?*

One of the factors that can determine the immunogenicity mechanisms against antibody treatment is the presence of high amount of antibody aggregates (Bartelds GM, et al. Development of antidrug antibodies against adalimumab and association with disease activity and treatment failure during long-term follow-up. JAMA, 2011; Rosenberg AS, et al. Effects of protein aggregates: an immunologic perspective. AAPS J, 2006), whose uptake can be carried out by antigen presenting cells or B cells through epitopes presentation. Clearly, pharmaceutical industry and regulatory agencies have been working to characterize the aggregates that seem mainly involved in the development of immunogenicity (Carpenter JF, et al. Overlooking subvisible particles in therapeutic protein products: gaps that may compromise product quality. J Pharm Sci, 2009). In line with these finding, we have studied antibodies aggregates by performing Dynamic light scattering (DLS) and Nanotracking analysis (NTA) of anti-IGF2R formulation from R&D System formulation (Appendix Fig S1A and B). Material and method description as well as the results are now included in the Appendix Supplementary Methods and in Figure S1 caption. In detail, size distribution of anti IGF2R aggregates was detected by DLS technique. We found two major peaks in the ranges of 10-20 nm (centered at 12.8nm \pm 3.77) and 180-200nm (centered at 196.8nm \pm 56), with an intensity percentage of 71.5% and the 28.5%, respectively. Saline solution did not shows notable peaks. NTA detection of anti IGF2R aggregate size distribution showed particle concentration in the main regions between 0 and 80 nm (One-way ANOVA, $p > 0.1$). Analysis of reactivity against polyclonal goat anti IGF2R antibody in plasma samples have not shown significant differences in untreated and treated mdx mice (One-way ANOVA, $p > 0.05$). Thus, the absence of antibodies against goat anti IGF2R in mdx treated mice suggests that the nano-sized aggregates of anti IGF2R are not immunogenic in these mice.

2. Could a mouse immune response to the goat polyclonal anti-IGF2R antibody complicate data interpretation?

The shortage of commercially available anti IGF2R antibody, for in vivo neutralization experiments, has compelled us to choose the polyclonal goat anti IGF2R from R&D System, on the basis of previously published data describing the injection of this antibody into murine models (Chen D, et al. A critical role for IGF-II in memory consolidation and enhancement. Nature, 2011). We adjusted the in vivo procedure and the antibody concentration to best fit for the mdx animal model and the intravenous rout of administration, also limiting the immune response risk by lying within the antibody concentration range shown in literature. Even if the engagement of an immune response cannot be completely ruled out, mainly arising either from the mouse against the goat antibody or from the own dystrophic inflammatory environment, we have observed a clear overall beneficial effect, supported by several and freestanding experimental evidences. In addition, during the experiments we have never observed adverse reactions, such as unspecific cardiac or respiratory symptoms or skin lesions, in line with previous results reporting the use of goat anti mouse antibodies (IL6, TNF alpha) (Stephen E., et al. Neutralization of tumor necrosis factor alpha reverses insulin resistance in skeletal muscle but not adipose tissue. Am J Physiol Endocrinol Metab, 2004; Pelosi L., et al. Functional and Morphological improvement of dystrophic muscle by interleukin 6 receptor blockade. EbioMedicine 2015). Nevertheless, we are aware that we need to develop appropriate antibodies exploiting chimeric or humanized antibodies or even peptide synthesis, sustained by solid knowledge of kinetic and biodistribution, to build further a safe and more feasible approach towards the applicability of this study as therapy.

3. How much of the anti-IGF2R antibody is present in the muscle of mdx mice after 9 weeks of treatment?

4. Is the treatment antibody at steady state in the muscle or do levels fluctuate with treatment?

We detected both light (23 kDa) and heavy (53.6 kDa) chain goat IgG in TAs and VMs from high and low dose anti-IGF2R injected mice. Untreated mdx and C57Bl murine muscles were used as controls. Ponceau staining was used for WB lane quantification. These data are now shown in Appendix Figure S1.

5. What is the proposed mechanism of IGF2R blockade using this antibody e.g. blocking the ligand-receptor interaction or receptor internalization and loss or some other mechanism(s)?

Details of proposed mechanism of IGF2R blockade are described in the new version of synopsis and through the discussion section of the mail text. However, we here summarize the main aspects of blockade machinery.

We found increased expression of IGF2R in dystrophic muscles and demonstrated its binding to CD20. Blockade of IGF2R facilitates IGF2-IGF1R interactions and activate CD20 phosphorylation promoting the entrance of Ca²⁺ ions in the sarcoplasm. STIM1 and ORAI1 are the two most prominent proteins involved in activation and regulation of store-operated Ca²⁺ entry (SOCE). STIM1 senses the Ca²⁺ concentration in cellular stores. Depletion of ER calcium results in the horizontal movement of STIM1 in the SR membrane, causing it to cluster and interact with the plasma membrane channel-forming protein ORAI1. This results in Ca²⁺ entry into the cell. The CD20 phosphorylation induced by IGF2R blockade in myoblasts decreases the interaction between CD20 and ORAI1 in store-depleted myoblasts, and this effect is increased in store-depleted myoblasts treated with anti-IGF2R. These data corroborate the hypothesis that CD20 interacts with ORAI1 in the muscle plasma membrane and that its phosphorylation promotes the interaction between ORAI1 and STIM1, which mediate Ca²⁺ release from intracellular calcium stores in the SR. Increasing levels of Ca²⁺ ions regulate calcineurin/CaMKII pathway and activate SERCA1 leading to increased force production and vasculature remodeling. SERCA activity has been reported to be reduced in dystrophic muscle. Blockade of IGF2R activated SERCA1 and enhanced SR Ca²⁺ uptake, promoting premature differentiation of myoblasts and correcting Ca²⁺ overload. CaMKII can activate the reuptake of Ca²⁺ ions in the SR regulating the SERCA1. The results of IGF2R blockade of myoblasts suggest that calcineurin inhibits CaMKII-mediated phosphorylation, and the inhibition of calcineurin increases phospho-CaMKII, which results in the stimulation of CaMKII-dependent cellular actions. Otherwise, the IGF2R blockade of mdx mice leads to negative regulation of CaMKII and activation of calcineurin. Moreover, calcineurin dephosphorylates nuclear factor of activated T-cells (NFAT) hereby regulating its nuclear localization and facilitate the increased expression of genes involved in myogenic program. Our results further support the idea that blockade of IGF2R increases intracellular Ca²⁺ of myoblasts activating cytoplasmic signaling cascades with opposite effects on calcineurin activity, and the net effect is NFAT dephosphorylation and translocation into the nucleus. Moreover, anti-IGF2R binding to domain 11 of IGF2R activates IGF2R-Gai2 interactions, prevents its interaction with IGF2 and leads to a decrease in the degradation. In turn, the increased bioavailability of IGF2 for IGF1R interactions leads to consequent IGF1R phosphorylation that recruits the PI3-K/Akt/mTOR signaling axis regulating skeletal muscle-specific genes associated with myogenic differentiation. The IGF1-induced pathway also increases Ca²⁺ influx via SOCE activation. PI3-K regulates the CD20 phosphorylation. Thus, blockade of IGF2R reestablishes the correct oscillating pattern of Ca²⁺ ion levels in the microenvironment of myofibrils, and exerts a protective role from degenerative mechanisms of dystrophic muscles. We provide evidence of physical and functional interactions between IGF2R and CD20. Perturbing this interaction with anti-IGF2R increases IGF bioavailability to IGF1R and reduces the intracellular Ca²⁺ concentration in dystrophic muscle cells, eventually resulting in an extremely significant amelioration of dystrophic muscle histology, vasculature defects and force performance.

Corresponding Author Name: Yvan Torrente

Manuscript Number: EMM-2019-11019-V2